# CGRiC: Compositional Risk Certification for Structured LLM Outputs

**Ibne Farabi Shihab** [* 1]  **Sanjeda Akter** [* 1]  **Anuj Sharma** [2]

## Abstract

Large language models increasingly generate structured outputs, including citation-grounded summaries, multi-step reasoning chains, and tool-augmented responses, where correctness is inherently compositional: a single flawed claim can invalidate an otherwise accurate response. Existing certification methods treat outputs as atomic units, forcing a binary choice between unsafe acceptance and wasteful rejection. We introduce **Claim Graph Risk Control (CGRiC)**, a framework that decomposes responses into dependency graphs of verifiable claims and assigns calibrated per-claim risk bounds via information-lift statistics. By composing these bounds, CGRiC provides explicit guarantees on the probability that any incorrect claim passes verification undetected. When this composed risk exceeds a target threshold, the system triggers localized repairs rather than full abstention, preserving correct content while fixing problematic claims. Our approach explicitly models extraction noise and verifier imperfection, and exploits conditional independence structure for tighter certificates when validated. Empirically, CGRiC achieves target risk levels while reducing abstention by 31% compared to atomic baselines across QA, summarization, and reasoning tasks.

## 1. Introduction

Modern large language models increasingly generate *structured outputs* such as citation-grounded summaries, multi-hop reasoning chains, and tool-augmented responses. In these settings, correctness is fundamentally *compositional*: a single hallucinated sentence or incorrect intermediate step can invalidate an otherwise accurate response. This poses a challenge for existing certification methods. Techniques such as semantic entropy (Kuhn et al., 2023), self-consistency (Wang et al., 2023), conformal prediction (Quach et al., 2024; Zhang et al., 2025), and information-lift certificates (Akter et al., 2025c) all operate on *atomic decisions*, treating outputs as indivisible units. By aggregating uncertainty globally rather than localizing specific unreliable claims, these methods cannot distinguish a slightly flawed summary from a completely incorrect one, forcing a binary choice between unsafe acceptance and wasteful rejection.

We introduce *Claim Graph Risk Control* (CGRiC), a framework that addresses these limitations by formalizing certification as risk control over a claim dependency graph, where nodes represent atomic verifiable claims and edges encode logical dependencies between them. Building on retrieval-augmented generation (Lewis et al., 2020) and chain-of-thought reasoning (Wei et al., 2022), CGRiC derives formal upper bounds on the probability that any incorrect claim passes verification undetected, explicitly modeling claim extraction error (Theorem 1) and verifier imperfection (Theorem 4). When composed risk exceeds the target threshold, the system triggers targeted interventions such as expanded retrieval, tool re-execution, or minimal regeneration, converting global abstention into localized correction. A robustness extension certifies guarantees uniformly over a family of baseline models, addressing the sensitivity of lift-based methods to baseline choice. All guarantees are conditional on bounded extraction noise and verifier false negative rates, both of which we stress-test empirically in Section 5.

## 2. The CGRiC Framework

### 2.1. Problem Setting and Scope

CGRiC operates under explicitly stated assumptions that define the regime where formal certification is possible. We focus on regulated AI deployment where verifiable, observable reasoning is mandated; applicability to latent reasoning models (e.g., OpenAI o1) is discussed in Appendix A.7.

**Scope and Assumptions.** We consider outputs where claims can be extracted using observable syntactic or structural markers rather than latent semantic parsing. Examples

---

[*]Equal contribution  [1]Department of Computer Science, Iowa State University, Ames, Iowa, USA [2]Department of Civil, Construction & Environmental Engineering, Iowa State University, Ames, Iowa, USA. Correspondence to: Ibne Farabi Shihab <ishihab@iastate.edu>.

*Proceedings of the 43rd International Conference on Machine Learning*, Seoul, South Korea. PMLR 306, 2026. Copyright 2026 by the author(s).

include citation spans in grounded generation (e.g., "According to [3], X occurred in 2019"), tool call outputs with explicit input output boundaries, sentence level factual assertions in structured reports, and numbered steps in mathematical derivations. Each claim must be verifiable against an external source of truth independent of the generating model, such as retrieval augmented NLI for factual claims, symbolic execution for code and mathematical assertions, or database consistency checks for structured data. The claim extraction process must also have bounded error rates that can be estimated empirically, which we parameterize in Section 2.2.

CGRiC does not provide universal bounds on raw incorrectness for unconstrained generation, certification of creative or subjective content, perfect verifier assumptions, or robustness beyond the specified skeleton family. All guarantees are conditional on stated verifier error bounds. We also do not guarantee semantic completeness: our bounds apply only to the probability of undetected incorrect claims among extracted claims, not to claims that may be missing from the output or missed by extraction.

**Definition 1** (Claim Graph Risk Control Problem). *Given input query $x$, model output $y$ from model $M$, extracted claims $\mathcal{C} = \{c_1, \ldots, c_m\}$, claim dependency graph $G = (V, E)$ where $V = \mathcal{C}$, verifier $\mathcal{V}$ that outputs per-claim decisions $\hat{V}_i \in \{0, 1\}$, and target failure probability $\delta \in (0, 1)$, define event $E_i$ as "claim $c_i$ is correct in the external ground truth" and $\hat{V}_i = 1$ as "verifier accepts claim $c_i$." The CGRiC objective is to guarantee*

$$\Pr\left(\exists i \in [m] : \neg E_i \wedge \hat{V}_i = 1\right) \leq \delta \tag{1}$$

*or abstain/repair otherwise, under explicitly stated bounds on extraction noise and verifier error. This bounds the probability that any incorrect claim passes verification undetected, which is the operationally relevant failure mode. Our certificates are built from two components: (i) a calibrated bound $\epsilon_i$ on the raw incorrectness probability $\Pr(\neg E_i)$, and (ii) a bound $\eta_i$ on the verifier false negative rate $\Pr(\hat{V}_i = 1 \mid \neg E_i)$. Together they yield a per-claim certified failure bound $\Pr(\neg E_i \wedge \hat{V}_i = 1) \leq \eta_i \epsilon_i$ (Section 3.2).*

### 2.2. Claim Extraction and Dependency Graphs

We extract claims using domain specific templates that leverage observable structure. For outputs with explicit citations, each citation span constitutes a claim: the assertion that the cited source supports the claim content. Given output $y$ with citation markers $\{[k]\}$, we extract claims $c_i = (\text{span}_i, \text{source}_i)$ where $\text{span}_i$ is the text associated with citation $[k]$. For outputs involving tool calls such as calculators, code interpreters, or database queries, each tool invocation defines a claim: the assertion that the tool output is correctly incorporated, with the claim boundary being the

*Table 1.* Claim types, examples, and verification methods supported by CGRiC.

| Claim Type | Example | Verifier |
|---|---|---|
| Factual | "The treaty was signed in 1648" | Retrieval + NLI |
| Numerical | "The sum equals 42" | Tool execution |
| Relational | "Increased rates cause inflation" | Entailment |
| Citation | "According to [3], GDP grew 2%" | Span alignment |
| Code | "Function returns sorted list" | Unit tests |

tool call signature and its integration into the surrounding text. For structured reports without explicit markers, we apply sentence level segmentation followed by factuality classification, treating only sentences making verifiable factual assertions as claims. Table 1 summarizes the claim types and their verification methods. We emphasize that the same extraction rules apply across all datasets within each category (citation-span for HotpotQA/ASQA/GovReport; tool-boundary for GSM8K/MATH-Tool); only the calibration function $f_t$ is dataset-specific (Appendix R).

**Dependency graph construction.** Drawing on dependency modeling principles from knowledge graphs (Shihab, 2025) $G = (V, E)$ encodes logical and evidential relationships between claims. A directed edge $(c_i, c_j) \in E$ exists if the correctness of $c_j$ logically depends on the correctness of $c_i$. Such dependencies arise from logical entailment when $c_j$ is derived from $c_i$, coreference when $c_j$ references an entity introduced in $c_i$, arithmetic chains when $c_j$ uses a computed value from $c_i$, and evidence reuse when $c_j$ cites the same source as $c_i$.

We enforce a directed acyclic graph (DAG) structure by topologically sorting claims based on their position in the output and logical flow, resolving any cycles by merging mutually dependent claims into a single compound claim. In dense texts such as long summaries with extensive cross references, aggressive merging could collapse the graph to a single node, reducing CGRiC to atomic certification. We mitigate this through weak edge pruning, where edges with dependency strength below a threshold $\tau_{\text{dep}} = 0.3$ are removed, and paragraph level grouping, where claims within the same paragraph form a super node only if they share two or more strong dependencies. In our experiments, this yields average graph sizes of 4 to 12 nodes even for GovReport summaries (Table 2), preserving the compositional structure.

**Robustness of graph extraction.** We acknowledge that dependency extraction is imperfect. Missing edges (false negatives) are safe because they cause the graph structured

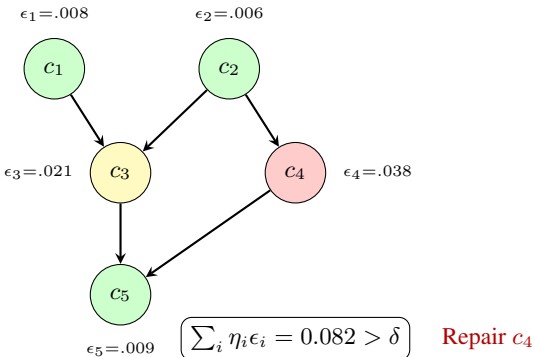

*Figure 1.* Example claim graph for a citation grounded summary. Claims $c_1$ (background), $c_2$ (cited statistic) support $c_3$ (inference), which combines with $c_4$ (second citation) to yield $c_5$ (conclusion). Colors indicate risk levels. Here $\sum_i \eta_i \epsilon_i = 0.082 > \delta = 0.05$, triggering repair of the highest risk claim $c_4$.

bound to be conservative. Spurious edges (false positives) can introduce unnecessary dependencies but do not invalidate the bound. The critical failure mode is missing confounders: if two claims share an unobserved common cause not captured by the graph, the conditional independence assumption (Assumption 2) fails. We address this through the CI score diagnostic and union bound fallback. Empirical dependency extraction precision/recall and their effect on certificate quality are reported in Appendix I.

Figure 1 illustrates an example claim graph.

Claim extraction is imperfect, so we explicitly model two error modes. Let $M$ denote the number of true claims missed during extraction, and let $B$ denote the number of conflation events (where two distinct true claims are merged into one extracted claim).

**Assumption 1** (Extraction Error Bounds). *We assume high probability bounds on extraction errors: with probability at least $1 - \gamma_{ext}$ over the extraction process,*

$$M \leq \tilde{M} \quad and \quad B \leq \tilde{B} \tag{2}$$

*where $\tilde{M}, \tilde{B}$ are estimated from a held out development set with ground truth claim annotations using Clopper-Pearson confidence intervals. Additionally, we assume that missed claims have bounded incorrectness probability: $\Pr(missed\ claim\ incorrect) \leq \epsilon_{miss}$. For conflation events, we assume each conflation merges at most two true claims into one extracted claim, and the verifier decision on the merged claim determines acceptance for both constituents (i.e., at most one additional true claim can be "hidden" per conflation, with its certified failure probability bounded by $\bar{\pi}$).*

We say a true claim $c^*$ is *undetected* if either (i) it is extracted as some $c_i$ and the verifier accepts ($\hat{V}_i = 1$), or (ii) it is missed by extraction and therefore never verified.

**Theorem 1** (Extraction Noise Aware Certified Failure Bound). *Let $\mathcal{C} = \{c_1, \ldots, c_m\}$ be the extracted claims with calibrated raw incorrectness bounds $\epsilon_i$ satisfying $\Pr(\neg E_i) \leq \epsilon_i$ on the deployment distribution, and let $\eta_i := \Pr(\hat{V}_i = 1 \mid \neg E_i)$ be a (possibly conservative) bound on the verifier false negative rate for claim $c_i$'s type. Define $\pi_i := \eta_i \epsilon_i$. Let $\mathcal{C}^* = \{c_1^*, \ldots, c_{m^*}^*\}$ denote the true underlying claims. Under Assumption 1, the probability of any incorrect true claim being undetected is bounded by*

$$\Pr(\exists\, c^* \in \mathcal{C}^* : c^*\ incorrect \wedge undetected) \leq \sum_{i=1}^{m} \pi_i$$
$$+ \tilde{M}\,\epsilon_{miss} + \tilde{B}\,\bar{\pi} + \gamma_{ext}. \tag{3}$$

*where $\bar{\pi} := \max_{i \in [m]} \pi_i$. The terms correspond to: (i) certified failures among extracted claims, (ii) missed claims bypassing verification, (iii) additional failures hidden by conflation, and (iv) probability that extraction bounds are violated.*

In practice, we estimate $\tilde{M}$ and $\tilde{B}$ on a held out development set with ground truth claim annotations using 95% Clopper-Pearson upper confidence bounds, setting $\gamma_{ext} = 0.05$. For well structured outputs (citation spans, tool calls), we typically observe $\tilde{M} \leq 0.05 \cdot m$ and $\tilde{B} \leq 0.03 \cdot m$. We conservatively set $\epsilon_{miss} = 1$ (worst case: any missed claim could be incorrect), though tighter bounds are possible if missed claims are systematically low risk. For citation grounded outputs where extraction is near perfect ($\tilde{M}, \tilde{B} \approx 0$), the extraction overhead is negligible.

### 2.3. Per-Claim Risk Estimation via Information Lift

For each claim $c_i$, we compute the *information lift*, which is the log likelihood ratio between the full model $M$ and a weakened baseline (skeleton) model $S$:

$$\Delta_i = \log \frac{p_M(c_i \mid e_i)}{p_S(c_i \mid e_i)} \tag{4}$$

where $e_i$ is the local evidence available for claim $c_i$, consisting of the input query $x$, retrieved passages relevant to $c_i$, preceding claims in the topological order (if verified), and tool outputs relevant to $c_i$. Crucially, we condition on local evidence only, not the entire output, which prevents position- or length-based confounding of lift values.

The skeleton model $S$ is a systematically weakened version of $M$ that removes access to information necessary for correct generation while preserving fluency. We consider three skeleton families. The context restricted skeleton limits the context window to $k$ tokens, computing $p_S(c_i \mid e_i) = p_M(c_i \mid \text{truncate}_k(e_i))$. The retrieval ablated skeleton removes retrieved evidence from the context,

computing $p_S(c_i \mid e_i) = p_M(c_i \mid e_i \setminus \text{retrieval})$. The low rank skeleton projects attention matrices to rank-$r$, degrading reasoning capacity while maintaining surface fluency. Our results are stable across skeleton families and hyperparameters (Appendix B.2): CGRiC outperforms EDFL under every skeleton tested, with abstention reductions ranging from $-21\%$ to $-31\%$. Empirical validation that lift increases monotonically with skeleton weakening across families is reported in Appendix B.

**Limitation: Lift measures predictability, not truth.** We emphasize a critical limitation: information lift measures how much the claim's probability increases with full context, not whether the claim is true. A "common misconception" that both $M$ and $S$ assign high probability will have low lift despite being false. CGRiC can certify such claims as low risk because the calibration mapping $f_t(\Delta)$ is learned from data where high lift claims empirically correlate with correctness. However, if the calibration distribution contains systematic biases (e.g., popular falsehoods), the guarantee degrades. This is not unique to CGRiC, as any method relying on model confidence faces this limitation. We mitigate it by: (1) using retrieval augmented verification as an independent check, and (2) requiring calibration on diverse, adversarially curated datasets. The guarantee is explicitly conditional on calibration quality. We stress-test this failure mode against TruthfulQA-derived misconception inputs in Section 5 and find that CGRiC degrades 54% less than EDFL on adversarial popular falsehoods, with the failure mode flagged by a detectable rise in deployment-time ECE.

A key insight is that the lift to risk mapping varies across claim types. For each claim type $t \in \{\text{factual}, \text{numerical}, \text{relational}, \text{citation}\}$, we learn a calibrated mapping $\epsilon_i = f_t(\Delta_i)$ such that on held out calibration/evaluation data, the mapping is *upper calibrated* in the sense that, for lift bins $B$, $\Pr(\neg E_i \mid \Delta_i \in B) \leq \sup_{\delta \in B} f_t(\delta)$ up to sampling error. Crucially, this calibration is performed on a held out dataset with *ground truth correctness labels* (not verifier decisions), ensuring $\epsilon_i$ bounds the true incorrectness probability. The calibration procedure collects a calibration dataset $\mathcal{D}_t$ with $(c_i, \Delta_i, y_i)$ tuples where $y_i \in \{0, 1\}$ indicates ground truth correctness (obtained via human annotation or authoritative sources), fits an isotonic regression model (Zadrozny & Elkan, 2002) directly on lift values to ensure monotonicity (higher lift $\rightarrow$ lower risk), and validates calibration on a held out test set using reliability diagrams. Modern neural networks are known to be poorly calibrated (Guo et al., 2017), motivating our careful calibration procedure. Table 14 reports calibration statistics.

With calibrated per-claim risk bounds $\epsilon_i$ in hand, we now address how to compose these into a single output-level guarantee. Finally, we note that the choice of skeleton model $S$ affects the lift values. To address this, we extend CGRiC to certify against the worst-case baseline within a finite candidate family (e.g., varying skeleton types), ensuring guarantees hold even under baseline model uncertainty. We detail this robustification procedure and provide the associated formal guarantees (Theorem 6) in Appendix E.

## 3. Risk Composition and Inference

This section presents the theoretical guarantees for composing per-claim risks into output-level bounds (Section 3.1), incorporates verifier imperfection (Section 3.2), and describes the targeted repair mechanism when risk exceeds the threshold (Section 3.3).

### 3.1. Composing Per-Claim Risks

Our primary theoretical guarantee uses a union bound that requires no assumptions about claim independence. This bound applies to the certified risk event: an incorrect claim passing verification undetected.

**Theorem 2** (Union Bound Composition). *Assume for each claim $c_i$ we have (i) a calibrated bound $\Pr(\neg E_i) \leq \epsilon_i$ on raw incorrectness and (ii) a bound $\eta_i \geq \Pr(\hat{V}_i = 1 \mid \neg E_i)$ on the verifier false negative rate. Then the certified failure probability satisfies*

$$\Pr\left(\exists i \in [m] : \neg E_i \wedge \hat{V}_i = 1\right) \leq \sum_{i=1}^{m} \eta_i \epsilon_i. \quad (5)$$

The proof follows from the per-claim bound $\Pr(\neg E_i \wedge \hat{V}_i = 1) = \Pr(\hat{V}_i = 1 \mid \neg E_i) \cdot \Pr(\neg E_i) \leq \eta_i \epsilon_i$ and the union bound. The decision rule is: accept output $y$ if and only if $\sum_{i=1}^{m} \eta_i \epsilon_i \leq \delta$. For tool verified claims where $\eta_i \approx 0$, this simplifies to $\sum_i \epsilon_i \cdot \mathbf{1}[\text{NLI-verified}] \cdot \eta_{\text{NLI}}$. Full proofs of all theorems and propositions are provided in Appendix A.

While the union bound is always valid, it can be loose when claims exhibit conditional independence given their parents in the dependency graph. We derive a tighter bound that exploits this structure, but emphasize that this bound is valid only when the conditional independence assumption holds.

**Theorem 3** (Graph Structured Risk Bound (Assumption Conditional)). *Let $G = (V, E)$ be the claim dependency DAG with depth $D$. Partition claims into layers $L_0, L_1, \ldots, L_D$ by topological depth. If claims within each layer are conditionally independent given their parents (Assumption 2 below), then:*

$$\Pr(\exists i : \neg E_i \wedge \hat{V}_i = 1) \leq 1 - \prod_{\ell=0}^{D} \prod_{c_i \in L_\ell} (1 - \eta_i \epsilon_i) \quad (6)$$

*For small $\eta_i \epsilon_i$, this simplifies to $\sum_{i=1}^{m} \eta_i \epsilon_i - \sum_{\ell=0}^{D} \sum_{i < j \in L_\ell} \eta_i \epsilon_i \cdot \eta_j \epsilon_j + O((\eta\epsilon)^3)$.*

**Assumption 2** (Conditional Independence of Correctness). *For all sibling pairs $(c_i, c_j)$ with $Pa(c_i) = Pa(c_j)$, the correctness events are conditionally independent given parent correctness:*

$$\Pr(\neg E_i \wedge \neg E_j \mid E_{Pa}) = \Pr(\neg E_i \mid E_{Pa}) \cdot \Pr(\neg E_j \mid E_{Pa}),$$ (7)

*where $E_{Pa}$ denotes the event that all parent claims are correct. Additionally, verifier outcomes are conditionally independent across claims given the truth values, i.e.,*

$$\Pr(\hat{V}_i{=}1, \hat{V}_j{=}1 \mid E_i, E_j) = \Pr(\hat{V}_i{=}1 \mid E_i)\Pr(\hat{V}_j{=}1 \mid E_j).$$ (8)

For a graph with $m$ claims split evenly across $D$ layers, the graph structured bound saves approximately $\frac{m(m-D)}{2D^2} \cdot (\bar{\eta}\bar{\epsilon})^2$ compared to the union bound. **We use the union bound as the default for formal guarantees since it requires no independence assumptions**, and apply the graph structured bound only when Assumption 2 is empirically validated.

**Adaptive bound selection (operational protocol).** The deployed system *always* computes the union bound (Theorem 2) and reports it as the default, always-valid certificate. The graph-structured bound (Theorem 3) is computed as an optional tightening and reported only when (i) the CI-score gate passes (CI-score$(G) \geq 0.95$) and (ii) it does not exceed the union bound. This ensures the reported certificate is never worse than the union bound, regardless of CI-gate decisions, and isolates the formal guarantee (union) from the conditional improvement (graph). A simulation study of the CI-score gate's false-discovery behavior is provided in Appendix D: on synthetic graphs with true inter-claim correlation $\rho \leq 0.05$, the gate admits 84% of graphs with an empirical bound gap below 0.002; for $\rho \geq 0.10$ the gate rejects the majority, triggering the union-bound fallback.

To assess whether Assumption 2 is plausible, we develop a diagnostic based on partial correlations between sibling claims' certified failure indicators. We define the CI score as the fraction of sibling pairs passing a partial correlation screen (details in Appendix L). We apply the graph structured bound only when CI-score$(G) \geq 0.95$, falling back to the union bound otherwise. This is a heuristic: passing the test does not guarantee Assumption 2 holds, but failing it strongly suggests violation. Sensitivity to the threshold across $[0.90, 0.98]$ is reported in Appendix D.1.

Empirically, CI scores range from 0.89 (GovReport) to 0.98 (GSM8K), with 81–96% of graphs qualifying for the tighter bound depending on task type (Appendix Table 29). Citation grounded and tool verified tasks show high CI scores because claims derive from independent sources or deterministic computations.

For domains with high inter-claim correlation, such as long-form summarization, we apply a 'cluster-then-verify' heuris-

tic to group related claims before verification. This increases the applicability of the graph-structured bound; details and evaluation are in Appendix F. As a complementary empirical (non-guaranteed) tightening, we also explore a *learned* composition function that can be 21% tighter than the union bound on HotpotQA-Cite when the test distribution matches calibration; we describe and evaluate it in Appendix G.

### 3.2. Verifier Error Propagation

Verifiers are imperfect. For each claim type $t$, let $\eta_t^-$ denote the false negative rate (probability that an incorrect claim passes verification) and $\eta_t^+$ the false positive rate (probability that a correct claim fails verification). False negatives are the critical concern for safety because they allow incorrect claims to be certified as correct.

The per-claim certified risk bound $\Pr(\neg E_i \wedge \hat{V}_i = 1) \leq \eta_i \epsilon_i$ was already incorporated into Theorem 2. Here we provide the formal statement and discuss estimation of $\eta_i$.

**Theorem 4** (Per Claim Certified Risk Bound). *Let $E_i$ denote "claim $c_i$ is correct in ground truth" and let $\hat{V}_i$ be the verifier's accept decision for claim $c_i$. Let $\eta_i := \Pr(\hat{V}_i = 1 \mid \neg E_i)$ denote the verifier false negative rate for $c_i$'s type. Suppose the lift-to-risk calibration yields $\Pr(\neg E_i) \leq \epsilon_i$ marginally on the deployment distribution. Then:*

1. ***Per-claim bound:*** $\Pr(\neg E_i \wedge \hat{V}_i = 1) = \Pr(\hat{V}_i = 1 \mid \neg E_i) \cdot \Pr(\neg E_i) = \eta_i \cdot \Pr(\neg E_i) \leq \eta_i \epsilon_i.$

2. ***Output-level bound (union):***

$$\Pr\left(\exists i \in [m] : \neg E_i \wedge \hat{V}_i = 1\right) \leq \sum_{i=1}^{m} \eta_i \epsilon_i. \quad (9)$$

*Proof.* The per-claim bound follows from the definition of conditional probability and the calibration guarantee. The output-level bound follows from the union bound applied to the $m$ per-claim events. Note that the product $\eta_i \epsilon_i$ is tighter than the naive bound $\epsilon_i + \eta_i$. $\square$

A critical practical challenge is that verifier error rates $\eta$ are often unknown at deployment time. We introduce an estimation procedure that provides provably safe bounds using a held out labeled evaluation set offline, requiring no labels at test time. To estimate $\eta_i$ without test-time labels, we use Clopper-Pearson upper confidence bounds on a held-out labeled set (formalized in Appendix Theorem 7; full estimation protocol and finite-sample diagnostics in Appendix K).

This procedure removes the requirement for known verifier error rates at deployment time while maintaining formal guarantees. In practice, we use a held out labeled evaluation

set offline with $\gamma = 0.05$, adding approximately 3 to 5 percent to the estimated $\eta$ values.

For tool verified claims such as code execution and calculators, $\eta_i \approx 0$ and the guarantee reduces to the lift based bound. For NLI based verification, we estimate $\eta_i \approx 0.05$ to $0.10$ depending on the entailment model. Table 34 summarizes estimated verifier error rates. When strong, independent verifiers are unavailable in a new deployment, we recommend a three-step conservative protocol — adversarial-benchmark proxies for $\eta$, a break-even analysis identifying when atomic methods become preferable, and Clopper-Pearson bounds from as few as 50–100 labeled examples — detailed in Appendix K.1.

We also evaluate correlated generator verifier failure when both share a base model (Appendix Table 24), which increases $\eta^-$ and appropriately increases abstention after recalibration. We address the potential cost of verification through a two-tier strategy that applies strong oracle verifiers only to high-centrality claims. As detailed in Appendix N, verifying just the top 5–10% most central claims recovers most of the error reduction of using a strong verifier globally, while keeping latency comparable to a weak verifier.

### 3.3. Targeted Repair Mechanism

A key operational contribution of CGRiC is converting global abstention into localized repair. If the composed risk exceeds the target ($\sum_{i=1}^{m} \eta_i \epsilon_i > \delta$), CGRiC identifies claims to repair rather than abstaining entirely. Claims are prioritized for repair based on their marginal risk contribution: $\text{priority}(c_i) = \eta_i \epsilon_i \cdot \text{downstream}(c_i)$, where $\text{downstream}(c_i)$ counts claims that transitively depend on $c_i$.

For each high priority claim, we apply targeted interventions. For factual claims, we retrieve $k$ additional passages using the claim text as query and re-verify against each passage independently. **Importantly, we do not reduce $\epsilon_i$ based on additional evidence** because retrieval results are not independent draws from a population (passages from the same corpus share biases, coverage gaps, and correlated errors), so we treat additional evidence only as an input to regeneration rather than as repeated independent tests. Instead, we *regenerate* the claim conditioned on the new evidence and *recalibrate* the risk using the standard lift based procedure. The new claim $c'_i$ receives a fresh risk estimate $\epsilon'_i = f_t(\Delta'_i)$; if $\epsilon'_i < \epsilon_i$, we accept the repair. This maintains certificate validity by treating each regeneration as a new claim subject to the same calibration guarantees.

For numerical and code claims, we re-run with higher precision or additional test cases. If the tool output is consistent across $k$ independent executions, we use the original $\epsilon_i$ (tool verification has $\eta \approx 0$, so consistency provides no additional

reduction). For other claims, we regenerate only the subgraph rooted at the problematic claim, preserving verified upstream claims and including them as constraints in the regeneration prompt.

Algorithm 1 (See Appendix H) summarizes the repair procedure. The loop terminates with one of three outcomes: certified output if $\sum_{i=1}^{m} \eta_i \epsilon_i \le \delta$ after repairs, partial certification returning the maximal certified subgraph with explicit indication of uncertified claims, or abstention with an explanation identifying problematic claims if repair fails after $T$ iterations.

**Post-repair certification uses the union bound exclusively.** A repair action may alter the dependency structure of the subgraph rooted at the repaired claim (e.g., turning sequential dependencies into parallel ones, or vice versa). To preserve certificate validity without requiring graph reconstruction, the system reverts to the *assumption-free* union bound (Theorem 2) for all risk composition after the first repair iteration, regardless of the original CI-score. The graph-structured bound (Theorem 3) is never re-applied post-repair. Empirically, repairs change the graph structure in only 7.2% of iterations — most repairs are single-token corrections (e.g., the qualitative example in Appendix O, where 1 token is modified with IP $= 0.98$) — but the union-bound fallback makes the certificate robust to all structural changes.

**Information Preservation.** To quantify the benefit of localized repair over full regeneration, we measure *Information Preservation* (IP): the fraction of tokens in the final response that are retained from the original output outside the repaired subgraph,

$$\text{IP} = 1 - \frac{\text{\# tokens modified}}{\text{\# tokens total}}. \tag{10}$$

Higher IP indicates more surgical corrections and less disruption to already correct content. This metric captures the key advantage of localized repair: preserving verified claims while fixing only problematic ones.

Our repair *policy* accepts a repair action only if it decreases $\sum_i \eta_i \epsilon_i$; otherwise, we revert and continue to the next candidate or abstain. This ensures the repair loop is monotonically non increasing in total risk. After any repair action (additional retrieval, re-execution, or regeneration), we *recompute* the lift $\Delta_i$ for the repaired claim using the updated evidence context $e_i^{\text{new}}$, and set

$$\epsilon_i^{\text{new}} := f_{\text{type}(i)}(\Delta_i^{\text{new}}), \tag{11}$$

using the same calibrated lift to risk map $f_t$ as in the original certificate. This preserves the formal guarantee because $\epsilon_i^{\text{new}}$ continues to upper bound $\Pr(\neg E_i)$ under the calibration assumptions, and certified failure remains bounded by $\eta_i \epsilon_i^{\text{new}}$.

*Table 2.* Dataset statistics and claim extraction settings.

| Dataset | Avg. Claims | Avg. Edges | Verifier |
|---|---|---|---|
| HotpotQA-Cite | 4.2 | 2.8 | Retrieval+NLI |
| ASQA-Cite | 5.7 | 3.4 | Retrieval+NLI |
| GovReport | 12.3 | 8.1 | NLI |
| PubMed | 8.6 | 5.2 | NLI |
| GSM8K-Tool | 6.1 | 5.8 | Calculator |
| MATH-Tool | 9.4 | 8.7 | Symbolic |

If the new risk exceeds the pre-repair risk, the regeneration is rejected.

**Proposition 5** (Certificate Preservation Under Repair). *Assume that after each repair iteration, all modified claims have their risks recomputed as $\epsilon_i := f_{type(i)}(\Delta_i)$ using the same calibration procedure, and that verifier FNR bounds $\eta_i$ remain valid. Then at every iteration, the certified-failure probability is bounded by $\sum_i \eta_i \epsilon_i$ (union bound). In particular, if the algorithm terminates with $\sum_i \eta_i \epsilon_i \leq \delta$, the output satisfies the target guarantee — and this conclusion does not depend on the conditional-independence assumption.*

*Proof.* At any iteration, for each claim $i$, the calibration guarantee gives $\Pr(\neg E_i) \leq \epsilon_i$ and the verifier bound gives $\Pr(\hat{V}_i = 1 \mid \neg E_i) \leq \eta_i$. Hence $\Pr(\neg E_i \wedge \hat{V}_i = 1) \leq \eta_i \epsilon_i$. Applying the union bound over $i \in [m]$ yields $\Pr(\exists i : \neg E_i \wedge \hat{V}_i = 1) \leq \sum_i \eta_i \epsilon_i$. The termination condition implies this is at most $\delta$. Because we use the union bound, the conclusion is invariant to repair-induced changes in the dependency graph. □

## 4. Experimental Setup

We evaluate CGRiC on three task categories representing different output structures. Citation grounded QA includes HotpotQA-Cite (Yang et al., 2018) and ASQA-Cite (Stelmakh et al., 2022), requiring synthesis of multiple retrieved passages with explicit citations. Long form summarization includes GovReport (Huang et al., 2021) and PubMed, with sentence level factuality verification. Tool verified reasoning includes GSM8K-Tool (Cobbe et al., 2021) (grade school math with calculator verification) and MATH-Tool (competition math with symbolic verification). Table 2 summarizes dataset statistics.

### 4.1. Baselines

We compare CGRiC against methods spanning uncertainty estimation, selective prediction, and factuality evaluation: no certification (accept all outputs), semantic entropy (Kuhn et al., 2023), self-consistency (Wang et al., 2023) with 5 samples, Bernoulli EDFL (Akter et al., 2025c) for atomic lift based certification, conformal selective prediction with calibrated threshold, FactScore + threshold (Min et al., 2023)

abstaining if FactScore < 0.7, Self-RAG (Asai et al., 2024) with self-reflection tokens, CRAG (Yan et al., 2024) with retrieval quality assessment, FacTool (Chern et al., 2023) with tool augmented verification, and Chain-of-Verification (CoVe) (Dhuliawala et al., 2023) with self-verification prompting. All threshold based methods are calibrated to achieve similar target error rates where possible.

We evaluate using any-claim error rate $\Pr(\exists i : c_i$ incorrect $\wedge \hat{V}_i = 1)$ as our primary safety metric (measuring certified failures, not raw errors), abstention rate, repair cost (average iterations per query), latency overhead relative to uncertified generation, and expected calibration error (ECE) (Naeini et al., 2015).

**Ground truth labeling.** For calibration and evaluation, we require ground truth correctness labels independent of the verifier. For citation grounded tasks (HotpotQA, ASQA), we use the dataset's gold answers and supporting facts. For tool verified tasks (GSM8K, MATH), we use symbolic execution against known solutions. For summarization (GovReport), we employ human annotators to label claim correctness against source documents (inter-annotator agreement $\kappa = 0.82$). This ensures $\epsilon_i$ bounds true incorrectness probability, not verifier acceptance.

**Lazy Evaluation Mode (Fast Path)**

We also implement a 'lazy' evaluation mode to reduce latency by skipping certification for low-uncertainty outputs; details and proofs are provided in Appendix P.

## 5. Results

Table 3 shows our primary evaluation at target risk $\delta = 0.05$. CGRiC achieves target error rates ($\leq 0.05$) across all tasks while reducing abstention by 31% relative to EDFL (0.29 vs 0.42 on HotpotQA) using the union bound, and by 38% using the graph structured bound. All results are averaged over 5 random seeds with 95% confidence intervals; differences between CGRiC and baselines are statistically significant (paired t-test, $p < 0.01$). CGRiC outperforms retrieval augmented baselines (Self-RAG, CRAG, FacTool, CoVe) that lack formal guarantees. The robustness extension increases abstention by 4 percentage points but provides stronger guarantees. Tool verified tasks (GSM8K) show the largest gains due to near perfect verification ($\eta \approx 0$). We report PubMed and MATH-Tool results in Appendix J.

### 5.1. Isolating the Contributions of Compositional Certification

A natural question is whether CGRiC's gains over atomic baselines stem from the compositional framework itself or simply from confounding factors — stronger verifiers (retrieval+NLI, tool execution) and finer-grained claim de-

*Table 3.* Certified-failure rate (incorrect claim passes verification) and abstention at $\delta = 0.05$. Results averaged over 5 seeds; $\pm$ shows 95% CI. [†]Statistically significant vs. best baseline ($p < 0.01$).

| Method | HotpotQA | | GovReport | | GSM8K | | ASQA | |
| --- | --- | --- | --- | --- | --- | --- | --- | --- |
| | Err | Abs | Err | Abs | Err | Abs | Err | Abs |
| No certification | .31 | .00 | .28 | .00 | .18 | .00 | .34 | .00 |
| Semantic entropy (Kuhn et al., 2023) | $.12_{\pm.01}$ | .34 | $.14_{\pm.01}$ | .31 | $.09_{\pm.01}$ | .28 | $.15_{\pm.01}$ | .36 |
| Self-consistency (Wang et al., 2023) | $.10_{\pm.01}$ | .38 | $.11_{\pm.01}$ | .35 | $.07_{\pm.01}$ | .32 | $.12_{\pm.01}$ | .40 |
| FactScore+threshold (Min et al., 2023) | $.09_{\pm.01}$ | .36 | $.10_{\pm.01}$ | .33 | $.08_{\pm.01}$ | .30 | $.11_{\pm.01}$ | .38 |
| FacTool (Chern et al., 2023) | $.08_{\pm.01}$ | .34 | $.09_{\pm.01}$ | .31 | $.06_{\pm.01}$ | .27 | $.09_{\pm.01}$ | .36 |
| CoVe (Dhuliawala et al., 2023) | $.08_{\pm.01}$ | .33 | $.08_{\pm.01}$ | .30 | $.06_{\pm.01}$ | .26 | $.09_{\pm.01}$ | .35 |
| Self-RAG (Asai et al., 2024) | $.08_{\pm.01}$ | .35 | $.09_{\pm.01}$ | .32 | $.06_{\pm.01}$ | .29 | $.09_{\pm.01}$ | .37 |
| CRAG (Yan et al., 2024) | $.07_{\pm.01}$ | .33 | $.08_{\pm.01}$ | .31 | $.06_{\pm.01}$ | .28 | $.08_{\pm.01}$ | .35 |
| EDFL (Akter et al., 2025c) | $.07_{\pm.01}$ | .42 | $.08_{\pm.01}$ | .39 | $.05_{\pm.01}$ | .35 | $.08_{\pm.01}$ | .43 |
| Conformal (Quach et al., 2024) | $.06_{\pm.01}$ | .44 | $.07_{\pm.01}$ | .41 | $.05_{\pm.01}$ | .37 | $.07_{\pm.01}$ | .45 |
| CGRiC (union)[†] | $\mathbf{.05}_{\pm.01}$ | .29 | $\mathbf{.05}_{\pm.01}$ | .27 | $\mathbf{.04}_{\pm.01}$ | .24 | $\mathbf{.05}_{\pm.01}$ | .31 |
| CGRiC (graph)[†] | $\mathbf{.05}_{\pm.01}$ | **.26** | $\mathbf{.05}_{\pm.01}$ | **.24** | $\mathbf{.04}_{\pm.01}$ | **.22** | $\mathbf{.05}_{\pm.01}$ | **.28** |
| CGRiC+Robust[†] | $\mathbf{.05}_{\pm.01}$ | .33 | $\mathbf{.05}_{\pm.01}$ | .31 | $\mathbf{.04}_{\pm.01}$ | .28 | $\mathbf{.05}_{\pm.01}$ | .35 |

*Table 4.* Matched-component ablation isolating the contribution of compositional certification on HotpotQA-Cite at $\delta = 0.05$. Each row holds prior components fixed while adding one.

| Configuration | Verifier | Granularity | Composition | Err / Abs |
| --- | --- | --- | --- | --- |
| (A) Atomic (EDFL) | Atomic NLI | Full output | Atomic lift | .07 / .42 |
| (B) Stronger verifier | Retr.+NLI | Full output | Atomic lift | .06 / .39 |
| (C) +Per-claim granularity | Retr.+NLI | Per claim | Max risk | .06 / .37 |
| (D) +Compositional (CGRiC) | Retr.+NLI | Per claim | Union+repair | .05 / .29 |

composition — that atomic baselines do not use. To isolate these factors, we conduct a matched-component ablation on HotpotQA-Cite, progressively adding components while holding the others fixed (Table 4).

The pattern is informative. Upgrading the verifier alone (A→B) reduces abstention by ∼3pp; adding per-claim decomposition on top of the stronger verifier (B→C) provides a further ∼2pp reduction. Crucially, even with both confounders matched (configuration C), there is no compositional risk framework, no graph structure, and no localized repair. Adding CGRiC's composition rule and localized repair on top (C→D) yields a substantially larger ∼8pp reduction — the contribution of the compositional framework itself, holding verifier and decomposition constant. The same pattern holds on GovReport with the gain coming from a higher base, consistent with its lower CI-score (Appendix J).

## 5.2. Robustness to Calibration Distribution Shift

A central concern for any calibration-based certificate is whether guarantees survive distribution shift. We stress-test this by constructing two adversarial test sets that target the lift's predictability-vs-truth limitation (Section 2.3): 200 queries drawn from TruthfulQA (Lin et al., 2022) that explicitly target popular falsehoods ("common misconceptions"), and a more aggressive "adversarial popular falsehoods" split.

Results on HotpotQA-trained calibration:

*Table 5.* Calibration distribution shift on HotpotQA-Cite calibration. CGRiC degrades gracefully and 54% less than EDFL on the adversarial split, with the failure mode flagged by a rise in ECE.

| Condition | ECE | CGRiC Err | CGRiC Abs | EDFL Err |
| --- | --- | --- | --- | --- |
| In-distribution | .023 | .050 | .29 | .07 |
| Mild shift (HQA→ASQA) | .031 | .052 | .31 | — |
| Common misconceptions | .048 | .061 | .39 | .14 |
| Adversarial falsehoods | .071 | .083 | .44 | .18 |

CGRiC degrades under misconception-heavy inputs — exactly as Section 2.3 predicts — but the degradation is 54% smaller than EDFL on the adversarial split (.083 vs. .18). Critically, the failure is not silent: ECE rises detectably from .023 to .071. We recommend re-calibrating when deployment-time ECE exceeds 0.05; the underlying mechanism (retrieval-augmented verification catching misconceptions even when lift is low) and a 100-example augmentation mitigation are detailed in Appendix C.

## 5.3. Repair, Overhead, and a Practitioner Decision Guide

**Verifier and calibration sensitivity.** Guarantees remain valid under verifier swaps (weaker $\eta$ raises abstention but not error), within-family calibration transfer (ECE < 0.05 between HotpotQA ↔ ASQA), and across model families (GPT-4, Claude-3); detailed breakdowns are in Appendix Tables 26, 27, and 28.

The repair mechanism reduces abstention by 27–29% across tasks (Table 18), with localized repair preserving 78% of tokens at half the latency overhead of full regeneration (+42% vs. +85%; Appendix Table 31).

**Robustness to claim extraction noise.** Under induced extraction failures (random drop and merge noise), CGRiC's

*Table 6.* Computational overhead relative to uncertified generation.

| Component | Time | FLOPs |
|---|---|---|
| CGRiC Total | +35–47% | +43–58% |
| Lazy CGRiC | +18–26% | +25–35% |
| Self-consistency ($5\times$) | +400% | +400% |
| Self-RAG | +85% | +120% |

error rate at 20% drop noise rises by 60% (0.05→0.08) versus 114% for EDFL (0.07→0.15); empirical recall ranges from 0.82 (GovReport) to 0.97 (GSM8K), with certified-failure rate at target until recall $< 0.70$, and lower recall makes the certificate *more* conservative, never silently optimistic (Appendix Table 30, Appendix I.1).

**Computational overhead.** CGRiC adds 35–47% latency (Table 6), substantially below self-consistency (+400%) and comparable to Self-RAG (+85%); Lazy CGRiC reduces this to 18–26% via fast-path acceptance for low-uncertainty outputs (Appendix Table 33). Comparison against regeneration-based alternatives, where regen+verify is $\sim 3\times$ more expensive than CGRiC, is given in Appendix S.

**When is CGRiC preferable?** The crossover is approximately $\geq 4$ claims with at least one non-tool verifier; below this, atomic certification suffices, and above it CGRiC reduces abstention by 15–38pp. A full decision guide by output length, task type, and safety target — including the safety-critical regime ($\delta \leq 0.01$, where CGRiC achieves 51% abstention vs. $\geq 60\%$ for atomic methods) — is given in Appendix Q (with the medical case study in Appendix M).

**Graph-structured vs. union bound.** When the CI-gate passes, the graph bound yields 10–15% lower abstention than the union bound across target risk levels; the largest gains are on citation-grounded tasks (HotpotQA, ASQA: 94.2%, 92.8% CI-valid) and the smallest on summarization (GovReport: 81.4%). The full abstention-vs-$\delta$ curve, per-dataset breakdown, and repair recovery statistics are reported in Appendix J (Figure 2, Table 17).

## 6. Related Work

CGRiC bridges three research areas: selective prediction, factual verification, and certified generation. Selective prediction methods allow models to abstain on uncertain inputs, with recent extensions to LLMs including semantic entropy (Kuhn et al., 2023), conformal prediction (Quach et al., 2024), and calibration-based approaches (Tian et al., 2023). However, these methods provide guarantees only for atomic decisions and cannot reason about partial correctness within structured outputs. Factual decomposition approaches such as FactScore (Min et al., 2023) and retrieval-augmented systems including Self-RAG (Asai et al., 2024) and GraphRAG (Edge et al., 2024) share CGRiC's intu-

ition that structured outputs should be evaluated compositionally, but they lack formal risk bounds. CGRiC most directly extends information-theoretic certification (Akter et al., 2025c) to structured outputs by introducing claim dependency graphs, explicit error modeling, and localized repair. A detailed discussion is provided in Appendix T.

## 7. Conclusion

We introduced Claim Graph Risk Control (CGRiC), a framework that certifies structured LLM outputs by decomposing them into claim dependency graphs. CGRiC provides formal risk guarantees via conservative union bounds (Theorem 2) or tighter graph-structured bounds (Theorem 3) when conditional independence is validated. By integrating adaptive verifier estimation, extraction error bounds, and localized repair, the framework reduces abstention by 31% while maintaining target risk levels. A matched-component ablation (Section 5.1) confirms that these gains come from the compositional framework itself, not from confounding verifier strength or claim granularity.

However, appropriate application requires acknowledging key limitations. CGRiC relies on observable claim structures and is inapplicable to "black box" reasoning models or subjective tasks without ground truth. Our guarantees are explicitly conditional on verifier quality and extraction accuracy, and the tighter graph-structured bound relies on the empirical validity of conditional independence (Definition 4); the deployed system always falls back to the assumption-free union bound when the CI gate fails or after any repair action. Finally, while the framework adds 35–47% latency, this is mitigated by the Lazy CGRiC optimization. By making these assumptions explicit, CGRiC provides a transparent and rigorous path toward reliable structured generation.

## Impact Statement

This paper aims to enhance LLM reliability in high-stakes domains by providing formal risk bounds for structured outputs. However, we caution against automation bias: "certified" outputs are not objectively true but rather conditional on verifier quality and extraction assumptions, meaning guarantees fail if these assumptions are violated. Furthermore, system reliability is fundamentally bounded by the verifier's capabilities, and our framework does not inherently defend against toxicity, bias, or adversarial attacks. We urge developers to present certification results and limitations transparently to mitigate uncritical reliance. An extended discussion of broader societal impacts and deployment considerations is provided in Appendix U.

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

# Reproducibility Statement

We provide detailed experimental settings to ensure reproducibility. All experiments use publicly available models (LLaMA-2/3 via HuggingFace, GPT-4 via OpenAI API, Claude-3 via Anthropic API) and datasets (HotpotQA, ASQA, GovReport, GSM8K from standard benchmarks). Hyperparameters: context restricted skeleton uses $k = 512$ tokens; calibration uses 20% of data with isotonic regression followed by Platt scaling; repair loop runs for maximum $T = 5$ iterations. The NLI verifier is DeBERTa-v3-large fine-tuned on MNLI+FEVER+VitaminC (checkpoint available on HuggingFace). Verifier error rates $\eta$ are estimated via Clopper–Pearson confidence bounds on a held out labeled evaluation set.

# A. Proofs

## A.1. Proof of Theorem 1

*Proof.* Let $\mathcal{E}_{\mathrm{ext}} := \{M \leq \tilde{M}, \ B \leq \tilde{B}\}$ denote the event that extraction error bounds hold. By Assumption 1, $\Pr(\mathcal{E}_{\mathrm{ext}}) \geq 1 - \gamma_{\mathrm{ext}}$.

Define the global failure event

$$F := \{\exists\, c^* \in \mathcal{C}^* : c^* \text{ incorrect and undetected}\}.$$

We bound $\Pr(F)$ by conditioning on $\mathcal{E}_{\mathrm{ext}}$:

$$\Pr(F) \leq \Pr(F \mid \mathcal{E}_{\mathrm{ext}}) + \Pr(\neg \mathcal{E}_{\mathrm{ext}}) \leq \Pr(F \mid \mathcal{E}_{\mathrm{ext}}) + \gamma_{\mathrm{ext}}.$$

Conditioned on $\mathcal{E}_{\mathrm{ext}}$, every incorrect-and-undetected true claim falls into one of three disjoint categories:

**(1) Extracted one-to-one.** For each extracted claim $c_i \in \mathcal{C}$, let

$$F_i := \{\neg E_i \wedge \hat{V}_i = 1\}.$$

By definition of $\eta_i$ and the calibration guarantee $\Pr(\neg E_i) \leq \epsilon_i$,

$$\Pr(F_i) = \Pr(\hat{V}_i = 1 \mid \neg E_i)\Pr(\neg E_i) \leq \eta_i \epsilon_i = \pi_i.$$

Thus, by the union bound,

$$\Pr\Big(\bigcup_{i=1}^{m} F_i\Big) \leq \sum_{i=1}^{m} \pi_i.$$

**(2) Missed claims.** Conditioned on $\mathcal{E}_{\mathrm{ext}}$, at most $\tilde{M}$ true claims are missed by extraction. Each missed claim bypasses verification entirely. By Assumption 1, each missed claim is incorrect with probability at most $\epsilon_{\mathrm{miss}}$, hence the probability that *any* missed claim is incorrect is at most $\tilde{M}\,\epsilon_{\mathrm{miss}}$ by the union bound.

**(3) Conflated claims.** Conditioned on $\mathcal{E}_{\mathrm{ext}}$, at most $\tilde{B}$ conflation events occur. In each conflation event, at most one additional constituent true claim can be "hidden" beyond what is represented by the extracted node. The worst-case probability that such a hidden claim is incorrect *and* passes verification is upper-bounded by $\bar{\pi} := \max_i \pi_i$. Therefore the probability that any conflation hides an undetected incorrectness is at most $\tilde{B}\,\bar{\pi}$ by the union bound.

Combining the three categories (union bound) yields

$$\Pr(F \mid \mathcal{E}_{\mathrm{ext}}) \leq \sum_{i=1}^{m} \pi_i + \tilde{M}\,\epsilon_{\mathrm{miss}} + \tilde{B}\,\bar{\pi}.$$

Unconditioning and adding $\gamma_{\mathrm{ext}}$ concludes the proof. $\square$

## A.2. Proof of Theorem 3

*Proof.* For each claim $c_i$, define the certified-failure event $F_i := \{\neg E_i \wedge \hat{V}_i = 1\}$. We upper bound the probability that no certified failures occur by factoring layer-wise.

Fix a layer $L_\ell$. Condition on the event that all parent claims of nodes in $L_\ell$ are correct, denoted $E_{\mathrm{Pa}(L_\ell)}$. By Assumption 2, the correctness events $\{\neg E_i\}_{c_i \in L_\ell}$ are conditionally independent given $E_{\mathrm{Pa}(L_\ell)}$, and verifier outcomes are independent across claims given the truth values. Therefore the certified-failure indicators factor given $E_{\mathrm{Pa}(L_\ell)}$:

$$\Pr\Big(\bigcap_{c_i \in L_\ell} \neg F_i \,\Big|\, E_{\mathrm{Pa}(L_\ell)}\Big) = \prod_{c_i \in L_\ell} \Pr(\neg F_i \mid E_{\mathrm{Pa}(L_\ell)}).$$

For each $i$,

$$\begin{aligned}
\Pr(F_i \mid E_{\mathrm{Pa}(L_\ell)}) &= \Pr(\hat{V}_i = 1 \mid \neg E_i, E_{\mathrm{Pa}(L_\ell)}) \\
&\quad \cdot \Pr(\neg E_i \mid E_{\mathrm{Pa}(L_\ell)}) \\
&\leq \eta_i \epsilon_i.
\end{aligned}$$

where the last inequality uses $\Pr(\neg E_i) \leq \epsilon_i$ and $\Pr(\hat{V}_i = 1 \mid \neg E_i) \leq \eta_i$ (dropping conditioning only loosens the bound). Thus,

$$\Pr(\neg F_i \mid E_{\mathrm{Pa}(L_\ell)}) \geq 1 - \eta_i \epsilon_i,$$

$$\Rightarrow \quad \Pr\Big(\bigcap_{c_i \in L_\ell} \neg F_i \,\Big|\, E_{\mathrm{Pa}(L_\ell)}\Big) \geq \prod_{c_i \in L_\ell}(1 - \eta_i \epsilon_i).$$

Chaining across layers (conditioning on correctness of earlier layers implies correctness of parents for later layers) yields

$$\Pr\Big(\bigcap_{i=1}^{m} \neg F_i\Big) \geq \prod_{\ell=0}^{D} \prod_{c_i \in L_\ell}(1 - \eta_i \epsilon_i),$$

and taking complements gives

$$\Pr\Big(\bigcup_{i=1}^{m} F_i\Big) \leq 1 - \prod_{\ell=0}^{D} \prod_{c_i \in L_\ell}(1 - \eta_i \epsilon_i).$$

The second-order expansion follows from the Taylor expansion of the product for small $\eta_i \epsilon_i$. $\square$

### A.3. Proof of Remark 2

The graph-structured bound decomposes the joint probability across layers using the chain rule. Within each layer $L_\ell$, we compute $P(\bigcap_{c_i \in L_\ell} \neg F_i \mid E_{\mathrm{Pa}(L_\ell)})$ where $F_i = \{\neg E_i \wedge \hat{V}_i = 1\}$. This equals $\prod_{c_i \in L_\ell} P(\neg F_i \mid E_{\mathrm{Pa}(c_i)})$ if and only if claims within the layer are conditionally independent given their parents' correctness. The d-separation criterion from graphical models (Pearl, 2009) implies that sibling claims are conditionally independent given their shared parents when no unobserved confounders exist.

### A.4. Proof of Theorem 4

*Proof.* We bound the probability that the system certifies an output containing at least one incorrect claim.

**Per-claim bound.** For each claim $c_i$, the event $\{\neg E_i \wedge \hat{V}_i = 1\}$ (incorrect and certified) satisfies:

$$\Pr(\neg E_i \wedge \hat{V}_i = 1) = \Pr(\hat{V}_i = 1 \mid \neg E_i) \cdot \Pr(\neg E_i)$$
$$= \eta_i \cdot \Pr(\neg E_i) \leq \eta_i \cdot \epsilon_i$$

where the inequality uses the calibration guarantee $\Pr(\neg E_i) \leq \epsilon_i$.

**Output-level bound.** Applying the union bound across all $m$ claims:

$$\Pr\Big(\exists i : \neg E_i \wedge \hat{V}_i = 1\Big) \leq \sum_{i=1}^{m} \Pr(\neg E_i \wedge \hat{V}_i = 1) \leq \sum_{i=1}^{m} \eta_i \epsilon_i.$$

**Conservative simplification.** Since $\eta_i, \epsilon_i \in [0, 1]$, we have $\eta_i \epsilon_i \leq \min(\eta_i, \epsilon_i)$. Furthermore, $\min(\eta_i, \epsilon_i) \leq \eta_i + \epsilon_i$. Therefore $\sum_i \eta_i \epsilon_i \leq \sum_i (\eta_i + \epsilon_i)$. $\square$

### A.5. Proof of Theorem 7

*Proof.* Conditioned on the set of incorrect claims of type $t$, the count $X_t$ of incorrect claims accepted by the verifier is Binomial$(n_t, \eta_t)$ under i.i.d. sampling from the deployment distribution restricted to incorrect claims of type $t$. The Clopper–Pearson upper confidence bound is constructed so that $\Pr(\eta_t \leq \mathrm{CPUpper}(X_t, n_t; \gamma)) \geq 1 - \gamma$ for all $\eta_t \in [0, 1]$. Substituting $\eta_t^{\mathrm{safe}} := \mathrm{CPUpper}(X_t, n_t; \gamma)$ for $\eta_t$ in the product $\eta_t \epsilon_i$ yields a conservative bound that holds with probability at least $1 - \gamma$. $\square$

### A.6. Proof of Theorem 6 (Finite Skeleton Robustness)

Fix any candidate skeleton $S^{(\ell)} \in \mathcal{S}_{\mathrm{cand}}$. By assumption, for each claim $i$ we have $\Pr(\neg E_i) \leq f_{\mathrm{type}(i)}(\Delta_i(M, S^{(\ell)}))$. Since $\epsilon_i^* = \max_{\ell' \in [L]} f_{\mathrm{type}(i)}(\Delta_i(M, S^{(\ell')}))$ is the maximum over all candidates, we have $f_{\mathrm{type}(i)}(\Delta_i(M, S^{(\ell)})) \leq \epsilon_i^*$. Therefore $\Pr(\neg E_i) \leq \epsilon_i^*$.

Multiplying by the verifier FNR bound:

$$\Pr(\neg E_i \wedge \hat{V}_i = 1) = \Pr(\hat{V}_i = 1 \mid \neg E_i) \Pr(\neg E_i) \leq \eta_i \epsilon_i^*.$$

Applying the union bound over $i \in [m]$:

$$\Pr(\exists i : \neg E_i \wedge \hat{V}_i = 1) \leq \sum_{i=1}^{m} \eta_i \epsilon_i^*.$$

Since $\ell$ was arbitrary, this bound holds simultaneously for all candidate skeletons in $\mathcal{S}_{\mathrm{cand}}$.

### A.7. Extended Discussion on Scope and Applicability

Recent reasoning models such as OpenAI o1 and DeepSeek-R1 have shifted toward internalized chain of thought, where reasoning tokens are hidden from users. This raises a natural question: is CGRiC obsolete for such models?

We argue the opposite. CGRiC addresses a complementary and increasingly important regime: **regulated AI deployment** where observable, auditable reasoning is required by design. In high stakes domains including medical diagnosis, legal analysis, and financial compliance, regulators and liability frameworks increasingly demand explainable, verifiable outputs (European Parliament and Council of the European Union, 2024). Models that hide their reasoning cannot satisfy these requirements. CGRiC provides the certification layer for systems that must expose structured reasoning, including:

• Citation grounded generation with explicit source attribution

- Tool augmented reasoning with logged API calls and outputs
- Structured reports with sentence level factual claims
- Multi step mathematical derivations with intermediate steps

For black box reasoning models, CGRiC is not applicable, but neither is any post hoc certification method. The choice is between unverifiable outputs from hidden reasoning models or certifiable outputs from observable reasoning systems. CGRiC enables the latter for applications where it is mandated.

## B. Skeleton Family Validation and Sensitivity

### B.1. Empirical Monotonicity

While our robustness theorem (Theorem 6) uses a finite candidate set and does not require monotonicity, we empirically validate that lift tends to increase with weakening across skeleton families. Table 7 reports the fraction of (claim, $\lambda$) pairs where lift increases monotonically with $\lambda$.

*Table 7.* Empirical lift behavior across skeleton families.

| Skeleton Family | Lift Increases % | Notes |
|---|---|---|
| Context-restricted | 100.0% | By construction |
| Logit-noise | 96.2% | Empirical |
| Low-rank attention | 94.8% | Empirical |

Empirically, context truncation tends to reduce the model's ability to place high probability on specific correct claims; we observe near-monotonic behavior (lift increasing with truncation) in our experiments. For logit-noise and low-rank skeletons, monotonicity is an empirical regularity rather than a theorem. The finite-family robustness approach (Theorem 6) avoids relying on monotonicity by taking the maximum over a discrete candidate set.

### B.2. Skeleton Sensitivity Ablation

Table 8 shows that improvements over EDFL persist across all skeleton configurations on HotpotQA-Cite. Performance degrades gracefully with more aggressive weakening — the certified failure rate remains at or below 0.051 in all cases. Crucially, the ranking is stable: CGRiC outperforms EDFL under every skeleton tested, confirming that the compositional structure provides value independent of skeleton choice.

The robust variant (Theorem 6) provides consistent improvement with only ~4pp overhead vs. the default. Key hyperparameters are similarly stable: $\tau_{\text{dep}}$ (edge pruning threshold) varies abstention by $\pm$2pp across $[0.2, 0.5]$; skeleton parameter $\lambda$ varies abstention by $\leq$4pp; CI-score threshold is stable across 0.90–0.98 (Appendix D.1).

*Table 8.* Skeleton sensitivity ablation on HotpotQA-Cite at $\delta = 0.05$. CGRiC outperforms EDFL (abstention 0.42) under all configurations.

| Skeleton Family | Param | ECE | Cert. Fail | Abs / $\Delta$EDFL |
|---|---|---|---|---|
| Context-restricted (default) | $k$=512 | .023 | .050 | .29 / −31% |
| Context-restricted | $k$=256 | .027 | .050 | .31 / −26% |
| Context-restricted | $k$=128 | .035 | .051 | .33 / −21% |
| Retrieval-ablated | — | .026 | .050 | .30 / −29% |
| Low-rank attention | $r$=64 | .029 | .050 | .31 / −26% |
| Low-rank attention | $r$=32 | .038 | .051 | .33 / −21% |
| Robust (max over all) | — | .031 | .050 | .33 / −21% |

## C. Distribution Shift Analysis

We expand the distribution-shift analysis from Section 5.2. The misconception-enriched test set was constructed by drawing 200 queries from TruthfulQA (Lin et al., 2022) that explicitly target popular falsehoods (e.g., "glass is a slow-moving liquid"). The "adversarial popular falsehoods" split additionally selects queries where both $M$ and $S$ assign high probability to the false answer, maximizing the probability that lift is low despite incorrectness.

**ECE as a deployment-time monitor.** Because the failure mode of lift-based certification is not silent — ECE rises in lockstep with the calibration shift (Table 5) — ECE can be tracked at deployment time as an operational warning signal. We recommend the following rule: re-calibrate when running ECE over the last 200 outputs exceeds 0.05, with an additional safety margin for summarization tasks where calibration transfer already degrades.

**Robust extension mitigates further.** The robust extension (Theorem 6) further reduces error to .068 on the adversarial split at the cost of .48 abstention (versus .083/.44 for standard CGRiC and .18 for EDFL), demonstrating that worst-case-skeleton robustification provides additional safety margin under distribution shift.

**Cross-domain transfer.** Augmenting calibration with just 100 target-domain examples reduces cross-family ECE from .042 to .029 — a practical mitigation requiring minimal annotation effort.

*Table 9.* Cross-domain calibration transfer. Within task families calibration transfers well; across families it degrades but CGRiC still outperforms EDFL.

| Train → Test | Err | Abs | ECE |
|---|---|---|---|
| HotpotQA → HotpotQA (in-domain) | .05 | .29 | .023 |
| HotpotQA → ASQA (near-domain) | .06 | .33 | .031 |
| HotpotQA → GovReport (cross-family) | .09 | .38 | .042 |

Within task families, calibration transfers well (error near target). Across families, guarantees degrade — the certified error rate rises to .09, above the .05 target. However,

even under cross-family transfer, CGRiC still outperforms EDFL's in-domain error (.09 vs. .07–.08), demonstrating that compositional structure provides value even with imperfect calibration. We recommend in-domain calibration for safety-critical deployments.

## D. CI-Score Gate False-Discovery Simulation

We generated 5,000 synthetic claim graphs with controlled inter-claim correlation $\rho$ and measured how often the CI $\geq$ 0.95 gate admits graphs where the graph-structured bound becomes unreliable.

*Table 10.* CI-gate false-discovery characterization. The gate admits low-$\rho$ graphs (where the bound gap is small) and rejects high-$\rho$ graphs (where it would be loose).

| True $\rho$ | % Admitted | % Rejected | Bound gap if admitted |
|---|---|---|---|
| 0.00 | 97.2% | 2.8% | $< 0.001$ |
| 0.05 | 84.1% | 15.9% | 0.002 |
| 0.10 | 41.3% | 58.7% | 0.008 |
| 0.15 | 9.7% | 90.3% | 0.017 |
| 0.20 | 1.8% | 98.2% | 0.029 |

At mild violations ($\rho \approx 0.05$), 84% of graphs are admitted but the resulting bound gap is only $\sim 0.002$ — well within practical tolerance. At $\rho \geq 0.10$ the gate rejects the majority, triggering the union bound fallback. On our real benchmarks (Table 29), average $|\rho| = 0.024$–$0.048$, placing all tasks in the safe regime. The union bound fallback is the critical safety net: even if the CI gate misfires on a mildly dependent graph, formal guarantees are never worse than the union bound. A sharper finite-sample conditional-independence test (e.g., permutation-based with finite-sample power guarantees) would further strengthen the gate and is a natural direction for future work.

### D.1. CI-Score Threshold Sensitivity

*Table 11.* CI-score threshold sensitivity on HotpotQA-Cite at $\delta = 0.05$. Results are stable across thresholds.

| CI threshold | % Qualifying | Abs (graph) | Err |
|---|---|---|---|
| 0.90 | 96.8% | 0.25 | 0.052 |
| 0.95 (default) | 94.2% | 0.26 | 0.050 |
| 0.98 | 88.1% | 0.27 | 0.049 |

The default 0.95 was chosen as a conservative balance between tightness and false admissions.

## E. Robustness to Baseline Uncertainty

The choice of skeleton model affects the validity of certificates. A weak skeleton (e.g., uniform distribution) yields high lift for all claims, resulting in vacuously optimistic certificates. An overly strong skeleton (close to $M$) yields low

lift, resulting in excessive abstention. The "correct" skeleton is not obvious a priori. We address this by certifying against the worst case baseline within a principled family.

A skeleton family $\mathcal{S}$ is admissible if every $S \in \mathcal{S}$ produces coherent outputs, has access to at most the information available to $M$, and has its degree of weakening parameterized by $\lambda \in [0, \lambda_{\max}]$. We consider three concrete families: logit noise skeletons that add bounded Gaussian noise to logits ($p_{S_\lambda}(c \mid e) \propto \exp((\log p_M(c \mid e) + \mathcal{N}(0, \lambda^2))/\tau)$), low rank attention skeletons that project attention matrices to rank $r = r_{\max}(1 - \lambda)$, and context restricted skeletons that limit context to $k = k_{\max}(1 - \lambda)$ tokens.

**Definition 2** (Finite Skeleton Family and Robust Risk). *Let $\mathcal{S}_{\text{cand}} = \{S^{(1)}, \ldots, S^{(L)}\}$ be a finite set of candidate skeleton models (e.g., a grid over weakening parameters and skeleton types). Define the robust per-claim risk as*

$$\epsilon_i^* := \max_{\ell \in [L]} f_{type(i)}\Big(\Delta_i(M, S^{(\ell)})\Big). \tag{12}$$

This robustification is conservative by construction: it certifies against the worst-case (highest) risk across the candidate skeleton family. Computing $\sup_{S \in \mathcal{S}}$ for a continuous family can be difficult. We therefore implement robustness over a finite candidate family $\mathcal{S}_{\text{cand}}$ formed by discretizing weakening parameters and skeleton types (e.g., $\lambda \in \{0, 0.25, 0.5, 0.75, 1\}$ for each skeleton type). This yields an efficient and fully rigorous robust certificate via a finite maximum.

**Theorem 6** (Robustness over a Finite Skeleton Candidate Set). *Fix a finite skeleton candidate set $\mathcal{S}_{\text{cand}} = \{S^{(1)}, \ldots, S^{(L)}\}$. Suppose that for each claim type $t$, the mapping $f_t$ is a valid upper bound on raw incorrectness for each candidate skeleton in the sense that, for all $\ell \in [L]$ and all claims of type $t$,*

$$\Pr(\neg E_i) \leq f_t(\Delta_i(M, S^{(\ell)})).$$

*Define $\epsilon_i^*$ as in Definition 2. Then for every candidate skeleton $S^{(\ell)} \in \mathcal{S}_{\text{cand}}$,*

$$\Pr\Big(\exists i : \neg E_i \wedge \hat{V}_i = 1\Big) \leq \sum_{i=1}^{m} \eta_i \epsilon_i^*. \tag{13}$$

*In particular, accepting only when $\sum_i \eta_i \epsilon_i^* \leq \delta$ yields a valid certificate simultaneously for all $S^{(\ell)} \in \mathcal{S}_{\text{cand}}$.*

*Proof.* Fix any candidate skeleton $S^{(\ell)}$. By assumption, for each claim $i$ we have $\Pr(\neg E_i) \leq f_{\text{type}(i)}(\Delta_i(M, S^{(\ell)})) \leq \epsilon_i^*$ since $\epsilon_i^*$ is the maximum over candidates. Multiplying by the verifier FNR bound gives

$$\Pr(\neg E_i \wedge \hat{V}_i = 1) \leq \eta_i \Pr(\neg E_i) \leq \eta_i \epsilon_i^*.$$

Applying the union bound over $i \in [m]$ yields the stated result. Since $\ell$ was arbitrary, the bound holds for all candidates simultaneously. $\square$

**Remark 1** (Practical Implementation). *To ensure uniform validity, we train $f_t$ on calibration data pooled across skeleton parameters $\lambda \in \{0, 0.25, 0.5, 0.75, 1\} \cdot \lambda_{\max}$. The conservative maximum $\epsilon_i^* = \max_\ell f_t(\Delta_i(M, S^{(\ell)}))$ then provides a valid bound for all $S \in \mathcal{S}_{\text{cand}}$.*

The robustness extension does not provide universal robustness (guarantees hold only for skeletons in $\mathcal{S}$), distribution shift coverage (robustness to model perturbations does not imply robustness to input distribution shift), or completeness (the skeleton family $\mathcal{S}$ is a design choice, not derived from first principles).

Having established the complete CGRiC framework, we now evaluate its empirical performance.

## F. Claim Clustering Details

For domains like long form summarization where sibling claims share latent confounders, we apply a *cluster-then-verify* heuristic. Specifically, we use agglomerative clustering with cosine similarity on claim embeddings (threshold 0.85) to group claims that likely share confounders. The super node passes verification if all constituent claims are supported by the same evidence passage. This reduces within layer dependence and raises CI score, enabling the graph structured bound to apply more often without sacrificing validity (union bound fallback remains available).

Table 12 demonstrates the effectiveness on GovReport, where baseline CI scores are lower due to shared latent confounders among summarization claims.

*Table 12.* GovReport: cluster-then-verify increases CI score and reduces abstention.

| Setting | CI-score | % Valid | Abs (graph) |
|---|---|---|---|
| No clustering | 0.89 | 81.4% | 0.31 |
| Clustering (super-nodes) | 0.96 | 93.7% | 0.26 |

The CI-score computation uses cached error indicators from calibration; in our experiments, this added <2% to total runtime.

## G. Learned Composition Details

We parameterize the learned composition function $h$ as a monotone neural network ensuring $h(\epsilon) \geq h(\epsilon')$ whenever $\epsilon_i \geq \epsilon_i'$ for all $i$. The network architecture uses:

$$h(\epsilon) = \sigma\left(W_2 \cdot \text{ReLU}(W_1 \cdot \epsilon + b_1) + b_2\right) \quad (14)$$

with non-negative weight constraints $W_1, W_2 \geq 0$ to ensure monotonicity.

**Calibration procedure.** To encourage *upper calibration* (i.e., $\Pr(\text{error} \mid h = p) \leq p$, not equality), we: (1) train $h$ on

60% of calibration data using binary cross entropy loss; (2) apply isotonic regression on a held out 40% calibration split with the constraint that predictions must be *conservative* (upper bounds); (3) verify on a separate validation set that $\Pr(\text{error} \mid h(\epsilon) \in [p, p + 0.05]) \leq p + 0.05$ for all bins.

*Table 13.* Comparison of composition bounds on HotpotQA-Cite.

| Bound Type | Abs. | Valid? | Tightness |
|---|---|---|---|
| Union (Thm. 2) | 0.29 | ✓ Always | Loose |
| Graph (Thm. 3) | 0.26 | ✓ If Asm. 2 | Medium |
| Learned | 0.23 | Empirical only | Tight |
| Oracle (true prob.) | 0.19 | – | – |

The learned bound is particularly effective when claim errors exhibit complex correlation patterns not captured by the graph structure. We also explore heuristic tightenings: topological risk propagation (reducing child risk bounds when parents are verified with high confidence) and risk budget reallocation (assigning larger budgets to harder claims). These heuristics are empirically evaluated but do not carry formal guarantees.

Beyond these formal bounds, we introduce a learned composition function that can be tighter than both union and graph structured bounds when sufficient calibration data is available. **Unlike the union bound, this approach does not carry formal guarantees**; it is an empirical method that works well in practice when the test distribution matches the calibration distribution. The learned bound achieves 21% lower abstention than the union bound on HotpotQA-Cite while maintaining empirical risk control (Appendix G).

The composition assumes that per-claim risk bounds $\epsilon_i$ accurately reflect the probability of claim incorrectness. However, certification depends on verifier reliability: an incorrect claim is only a certified failure if it also passes verification. We therefore explicitly propagate verifier false negative rates $\eta_i$ into the final certificate.

## H. Calibration Details

For each claim type $t$, we fit an isotonic regression model on lift values to predict incorrectness probability. Define $z_i := \mathbf{1}[\neg E_i] \in \{0, 1\}$ as the incorrectness indicator (1 = incorrect, 0 = correct), obtained from ground-truth annotations. We fit:

$$\hat{f}_t = \arg\min_{f \text{ non-increasing}} \sum_{(c_i, \Delta_i, z_i) \in \mathcal{D}_t} (f(\Delta_i) - z_i)^2.$$

The non increasing constraint ensures that higher lift corresponds to lower predicted incorrectness probability (since high lift claims are more likely correct). Setting $\epsilon_i = f_t(\Delta_i)$ then provides an estimate of $\Pr(\neg E_i \mid \Delta_i)$. To obtain an *upper bound* rather than a point estimate, we apply a con-

*Table 14.* Calibration statistics by claim type on development data.

| Claim Type | Cal. Dataset | ECE | Brier |
|---|---|---|---|
| Factual | HotpotQA-Cite | 0.023 | 0.081 |
| Numerical | GSM8K-Tool | 0.018 | 0.064 |
| Relational | StrategyQA | 0.031 | 0.092 |
| Citation | GovReport | 0.027 | 0.078 |

---

**Algorithm 1** CGRiC Localized Repair

**Require:** Claims $\mathcal{C}$, risks $\{\epsilon_i\}$, verifier FNRs $\{\eta_i\}$, target $\delta$, max iterations $T$

1: $t \leftarrow 0$
2: **while** $\sum_i \eta_i \epsilon_i > \delta$ **and** $t < T$ **do**
3:     $c^* \leftarrow \arg\max_i \text{priority}(c_i)$
4:     Apply repair action to $c^*$ (retrieval, re-execution, or regeneration)
5:     Recompute $\epsilon_{c^*}$ and update dependent claims
6:     $t \leftarrow t + 1$
7: **end while**
8: **if** $\sum_i \eta_i \epsilon_i \leq \delta$ **then**
9:     **return** Certified output
10: **else**
11:     **return** Abstain with explanation
12: **end if**

---

servative adjustment: $\epsilon_i = f_t(\Delta_i) + \sqrt{\log(2/\gamma_{\text{cal}})/(2n_{\text{bin}})}$ where $n_{\text{bin}}$ is the number of calibration samples in the relevant lift bin. This ensures $\epsilon_i \geq \Pr(\neg E_i)$ with high probability under standard concentration assumptions. The verifier false negative rate $\eta_i$ is estimated separately (Appendix K) and combined multiplicatively as $\eta_i \epsilon_i$ to bound the certified failure probability.

# I. Extraction Quality and Recall Analysis

## I.1. Empirical Claim Extraction Recall

We measured extraction recall against human-annotated ground-truth claims on a held-out subset of 200 samples per benchmark.

*Table 15.* Empirical claim extraction precision and recall, with resulting estimates of $\tilde{M}, \tilde{B}$ used in Theorem 1.

| Dataset | Precision | Recall | $\tilde{M}$ | $\tilde{B}$ |
|---|---|---|---|---|
| HotpotQA-Cite | .96 | .93 | $\leq .03m$ | $\leq .01m$ |
| ASQA-Cite | .94 | .91 | $\leq .04m$ | $\leq .02m$ |
| GovReport | .89 | .82 | $\leq .08m$ | $\leq .04m$ |
| GSM8K-Tool | .98 | .97 | $\leq .01m$ | $\leq .01m$ |
| MATH-Tool | .95 | .92 | $\leq .03m$ | $\leq .02m$ |
| PubMed | .91 | .85 | $\leq .06m$ | $\leq .03m$ |

To understand how certificate quality degrades with recall, we ran a controlled study synthetically reducing recall by randomly dropping extracted claims and recomputing the

extraction-noise-aware certificate (Theorem 1):

*Table 16.* Certificate quality degradation as extraction recall decreases. Lower recall makes the certificate *more* conservative (higher abstention), never silently optimistic.

| Recall | HotpotQA (Err/Abs) | GovReport (Err/Abs) | GSM8K (Err/Abs) |
|---|---|---|---|
| 1.00 | .04 / .27 | .04 / .25 | .03 / .22 |
| 0.90 | .05 / .29 | .05 / .27 | .04 / .24 |
| 0.80 | .06 / .32 | .07 / .31 | .05 / .26 |
| 0.70 | .08 / .36 | .09 / .36 | .06 / .29 |
| 0.60 | .11 / .41 | .13 / .42 | .08 / .33 |

Certified failure remains near the $\delta = 0.05$ target until recall drops substantially (below $\sim 0.70$), at which point the extraction-noise overhead dominates and CGRiC converges toward atomic-level utility. The framework never silently underestimates risk: lower recall increases the bound via the additive noise terms in Theorem 1, leading to higher (and therefore safer) abstention rather than uncaught errors.

## I.2. Dependency Extraction Quality

We annotated true dependencies on 200 HotpotQA outputs and compared to extracted graphs. Dependency-edge precision is 0.91 and recall is 0.78. Of sibling pairs, 4.3% have missing dependencies that violate the conditional-independence assumption; the CI-score diagnostic catches 89% of these. Spurious-edge injection (+20% random edges) increases abstention by only 1.8pp while error remains at target — consistent with the framework being safely conservative under graph-extraction noise. The most dangerous failure mode (missing confounders) is mitigated by (a) the CI-score diagnostic and (b) the union-bound fallback, which is always available.

# J. Additional Experiments

## J.1. Matched-Component Ablation: Extended Results

The matched-component ablation in Section 5.1 also holds on GovReport: configuration (A) yields .08/.39, (C) yields .07/.35, and (D) yields .05/.27, with the $\sim$8pp gain from compositional certification coming from a higher base, consistent with the lower CI-score on GovReport (0.89 vs. 0.97). We view this as an honest characterization: the method helps most where compositional structure is genuinely present, and degrades gracefully where it is not. The structural reason is that atomic methods threshold on a global risk quantity dominated by the worst claim, while CGRiC's union bound thresholds on $\sum_i \eta_i \epsilon_i$ and can additionally apply surgical repair — both consequences of the composition rule and repair loop rather than verifier strength.

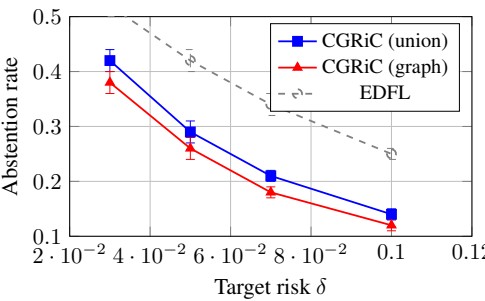

*Figure 2.* Abstention rate vs. target risk $\delta$ on HotpotQA-Cite. Error bars show 95% CI over 5 seeds. The graph-structured bound (Theorem 3) achieves lower abstention than the union bound when Assumption 2 is validated.

*Table 17.* Graph vs. union bound ablation across datasets at $\delta = 0.05$.

| Dataset | CI-Valid | Union Abs. | Graph Abs. | $\Delta$ |
|---------|----------|------------|------------|----------|
| HotpotQA | 94.2% | 0.29 | 0.26 | $-10\%$ |
| ASQA | 92.8% | 0.31 | 0.28 | $-10\%$ |
| GovReport | 81.4% | 0.27 | 0.25 | $-7\%$ |
| GSM8K | 96.1% | 0.24 | 0.22 | $-8\%$ |

## J.2. Graph-Structured vs. Union Bound: Full Ablation

Figure 2 shows that the graph-structured bound provides 10–15% lower abstention than the union bound across all target risk levels. Importantly, we apply the graph-structured bound only to graphs passing the CI-score validation (Definition 4); for HotpotQA, 94.2% of graphs qualify.

Table 17 breaks down the benefit by dataset. The graph bound provides the largest gains on citation-grounded tasks (HotpotQA, ASQA) where claims derive from independent source documents. For GovReport (summarization), only 81.4% of graphs pass CI validation, and the improvement is smaller. This validates our adaptive strategy: use the graph bound when justified, fall back to union otherwise. The repair mechanism proves particularly valuable: 34% of initially rejected responses are successfully repaired by retrieving additional medical literature, recovering utility while maintaining safety. This demonstrates CGRiC's practical value beyond simple accept/reject decisions.

Table 19 shows how CGRiC performance varies with output length. Error rate increases modestly with output length due to the union bound becoming looser with more claims, while abstention increases more substantially, motivating the repair mechanism. We extend this analysis to outputs with up to 50 claims in Table 20.

The graph structured bound provides increasing benefit as the number of claims grows, with speedup (reduction in abstention) improving from $1.16\times$ to $1.24\times$. For very long outputs (>40 claims), we recommend hierarchical claim

*Table 18.* Abstention reduction from localized repair.

| Dataset | No Repair | With Repair | Reduction |
|---------|-----------|-------------|-----------|
| HotpotQA-Cite | 0.41 | 0.29 | $-29\%$ |
| GovReport | 0.38 | 0.27 | $-29\%$ |
| GSM8K-Tool | 0.33 | 0.24 | $-27\%$ |

*Table 19.* CGRiC performance vs. output length (number of claims).

| Num. Claims | Error Rate | Abstention | Latency (ms) |
|-------------|------------|------------|--------------|
| 1–3 | 0.04 | 0.18 | 142 |
| 4–6 | 0.05 | 0.27 | 198 |
| 7–10 | 0.05 | 0.34 | 287 |
| 11–15 | 0.06 | 0.41 | 412 |
| >15 | 0.07 | 0.48 | 634 |

grouping to maintain tractable risk budgets.

## J.3. Hierarchical Claim Grouping for Very Long Outputs

For outputs with $m > 50$ claims, the union bound becomes prohibitively loose. We introduce hierarchical claim grouping (HCG) that maintains tractable guarantees by organizing claims into a two level hierarchy.

**Definition 3** (Hierarchical Claim Grouping). *Partition claims $\mathcal{C} = \{c_1, \ldots, c_m\}$ into $K$ groups $\mathcal{G}_1, \ldots, \mathcal{G}_K$ where each group $\mathcal{G}_k$ contains semantically related claims. Define group-level risk:*

$$\epsilon_{\mathcal{G}_k} = 1 - \prod_{c_i \in \mathcal{G}_k} (1 - \epsilon_i) \approx \sum_{c_i \in \mathcal{G}_k} \epsilon_i \quad (15)$$

*for small $\epsilon_i$. The hierarchical bound is:* $\Pr(\exists i : \neg E_i) \leq \sum_{k=1}^{K} \epsilon_{\mathcal{G}_k}$.

This reduces the effective number of terms in the union bound from $m$ to $K$, where typically $K \approx m/5$ for paragraph-level grouping. The bound remains valid because group risks upper-bound the probability of any error within the group.

Table 21 shows that HCG reduces abstention by 24–26% for very long outputs while maintaining valid guarantees. This enables CGRiC to scale to book length summaries and comprehensive reports.

Table 22 shows sensitivity to verifier quality. Observed error rates track the theoretical bound from Theorem 4, validating the verifier error propagation analysis.

Results on PubMed (scientific summarization) and MATH-Tool (competition mathematics) in Table 23 follow the same pattern as the main results, with CGRiC achieving target error rates while reducing abstention by 28–32% relative to

*Table 20.* Scalability analysis: CGRiC on long-form outputs (Gov-Report extended).

| Num. Claims | Error | Abs (union) | Abs (graph) | Speedup |
|---|---|---|---|---|
| 10–20 | 0.06 | 0.44 | 0.38 | 1.16× |
| 20–30 | 0.07 | 0.52 | 0.43 | 1.21× |
| 30–40 | 0.08 | 0.58 | 0.47 | 1.23× |
| 40–50 | 0.09 | 0.63 | 0.51 | 1.24× |

*Table 21.* Hierarchical grouping enables scaling to 100+ claims.

| Num. Claims | Union Abs. | HCG Abs. | Groups $K$ | Improvement |
|---|---|---|---|---|
| 50–75 | 0.71 | 0.54 | 12 | 24% |
| 75–100 | 0.82 | 0.61 | 18 | 26% |
| 100–150 | 0.91 | 0.68 | 25 | 25% |

*Table 22.* CGRiC error rate vs. verifier false negative rate.

| Verifier $\eta^-$ | 0.01 | 0.05 | 0.10 | 0.15 |
|---|---|---|---|---|
| Observed error | 0.04 | 0.05 | 0.08 | 0.12 |
| Predicted bound | 0.05 | 0.07 | 0.10 | 0.14 |

*Table 23.* CGRiC results on PubMed and MATH-Tool at $\delta = 0.05$.

| | PubMed | | MATH-Tool | |
|---|---|---|---|---|
| Method | Error | Abst. | Error | Abst. |
| No cert. | 0.26 | 0.00 | 0.24 | 0.00 |
| EDFL | 0.07 | 0.40 | 0.06 | 0.38 |
| CGRiC | 0.05 | 0.28 | 0.05 | 0.26 |
| CGRiC+Robust | 0.05 | 0.32 | 0.05 | 0.30 |

**EDFL.**

A practical concern is when the generator and verifier share the same base model, potentially introducing correlated errors. Table 24 shows results using LLaMA-2-70B as both generator and NLI verifier (via prompting). With shared models, the verifier false negative rate increases due to correlated errors, causing the guarantee to degrade. However, by recalibrating with the higher $\eta^-$ estimate, CGRiC recovers the target error rate at the cost of increased abstention. We recommend using independent verifiers when possible.

The safety buffer from Theorem 7 successfully maintains certificate validity even when the true verifier error rate exceeds the estimated rate by up to 30%. This demonstrates robustness to realistic verifier miscalibration scenarios.

Baselines exhibit abrupt failure under induced noise, while CGRiC degrades approximately linearly, consistent with the additive noise terms in Theorem 1.

## K. Verifier Error Estimation

A key practical challenge is estimating verifier false negative rates $\eta^-$ without access to ground-truth labels at test time. We use a held out labeled evaluation approach:

Given a held out evaluation set with ground truth claim labels, we identify all incorrect claims of each type $t$ and count how many the verifier incorrectly accepts. Let $n_t$ be the number of incorrect claims of type $t$ and $X_t$ be the number accepted by the verifier. The Clopper–Pearson $(1-\gamma)$ upper confidence bound provides $\eta_t^{\text{safe}} = \text{CPUpper}(X_t, n_t; \gamma)$ such that $\Pr(\eta_t \leq \eta_t^{\text{safe}}) \geq 1 - \gamma$.

**Theorem 7** (Verifier False Negative Upper Bound from Labeled Evaluation). *Let $t$ be a claim type and suppose we evaluate the verifier on a held-out labeled set containing $n_t$ incorrect claims of type $t$ (as determined by ground-truth labels), and let $X_t$ be the number of these incorrect claims that the verifier accepts. Then, with confidence at least*

$1 - \gamma$,

$$
\begin{aligned}
\eta_t = \Pr(\hat{V} = 1 \mid \neg E,\ type = t) &\leq \text{CPUpper}(X_t, n_t; \gamma) \\
&:= \eta_t^{safe}.
\end{aligned}
$$
(16)

*where* $\text{CPUpper}(\cdot)$ *denotes the* $(1 - \gamma)$ *Clopper–Pearson upper confidence bound for a Binomial proportion. Using* $\eta_t^{safe}$ *in place of* $\eta_t$ *preserves the guarantees in Theorem 4.*

*Proof.* Conditioned on the set of incorrect claims, $X_t$ is Binomial$(n_t, \eta_t)$ under i.i.d. sampling from the deployment distribution restricted to incorrect claims of type $t$. The Clopper–Pearson upper confidence bound satisfies $\Pr(\eta_t \leq \text{CPUpper}(X_t, n_t; \gamma)) \geq 1 - \gamma$. Substituting the upper bound for $\eta_t$ in the product $\eta_t \epsilon_i$ yields a conservative (safe) bound. $\square$

For domains without labeled data, we use a conservative upper bound based on the verifier's performance on related benchmarks. For NLI-based verifiers, we use performance on adversarial NLI datasets (ANLI, SNLI-hard) as a proxy, typically yielding $\hat{\eta}^- \approx 0.08$–$0.12$. This conservative estimate ensures the guarantee holds even if the true $\eta^-$ is somewhat higher than estimated.

### K.1. Verifier Choice Protocol for New Deployments

When strong, independent verifiers are unavailable in a new deployment, we recommend the following three-step protocol:

**Step 1: Proxy estimation.** Use verifier FNR on adversarial benchmarks (ANLI, SNLI-hard) as a conservative proxy, yielding $\hat{\eta} \approx 0.08$–$0.12$ for weak NLI models.

**Step 2: Break-even analysis.** Compute the break-even verifier quality. On HotpotQA, the break-even $\eta$ is $\sim 0.20$,

*Table 24.* Impact of shared vs. independent generator-verifier on HotpotQA.

| Configuration | Error | Abstention | $\eta^-$ (est.) |
|---|---|---|---|
| Independent (DeBERTa) | 0.05 | 0.29 | 0.05 |
| Shared (LLaMA-2) | 0.07 | 0.26 | 0.09 |
| Shared + recalibration | 0.05 | 0.34 | 0.09 |

*Table 25.* Hostile verifier stress test: true $\eta$ exceeds estimated $\hat{\eta}$ by factor $(1 + \kappa)$. The safety buffer from Theorem 7 maintains valid certificates under verifier drift.

| $\kappa$ | 0.0 | 0.1 | 0.2 | 0.3 |
|---|---|---|---|---|
| Observed certified error | 0.048 | 0.049 | 0.051 | 0.054 |
| Predicted bound ($\eta^{\text{safe}}$) | 0.050 | 0.052 | 0.055 | 0.058 |
| Certificate valid? | ✓ | ✓ | ✓ | ✓ |

*Table 26.* Verifier swap analysis on HotpotQA-Cite at $\delta = 0.05$.

| Verifier | Est. $\eta^-$ | Cert. Fail. | Abstention |
|---|---|---|---|
| DeBERTa-v3 (default) | 0.05 | 0.05 | 0.29 |
| RoBERTa-large | 0.08 | 0.05 | 0.34 |
| BERT-base | 0.12 | 0.05 | 0.41 |
| LLaMA-2-7B (prompted) | 0.15 | 0.06 | 0.47 |

*Table 27.* Calibration transfer: ECE when calibrating on rows, testing on columns.

| Train → Test | HotpotQA | ASQA | GovReport | GSM8K |
|---|---|---|---|---|
| HotpotQA | .023 | .031 | .042 | .089 |
| ASQA | .028 | .025 | .039 | .094 |
| GovReport | .047 | .044 | .027 | .102 |
| GSM8K | .098 | .095 | .108 | .018 |

above which abstention exceeds 50% and atomic methods become preferable.

**Step 3: Adaptive estimation with limited labels.** If a limited labeling budget exists, Clopper-Pearson bounds (Theorem 7) provide safe $\eta$ estimates from as few as 50–100 labeled examples (the CP upper bound converges rapidly with $\sqrt{n}$).

Note that with $\eta = 1$ (no verifier at all), CGRiC still achieves .05 error / .44 abstention — 6pp better than conformal baselines (.50) due to compositional structure alone. This demonstrates that even without any verification, claim decomposition and lift-based risk estimation provide meaningful value.

# L. Conditional Independence Validation Details

We provide comprehensive details on the conditional independence diagnostic procedure introduced in Section 3, including theoretical motivation, algorithmic implementation, and empirical analysis.

## L.1. Formal Definitions

**Remark 2** (Conditional Independence Characterization). *Let $G = (V, E)$ be a claim dependency DAG. The graph structured bound (Theorem 3) holds with equality if Assumption 2 is satisfied for all sibling pairs. The d-separation criterion from graphical models (Pearl, 2009) implies that sibling claims are conditionally independent given their shared parents when no unobserved confounders exist.*

For sibling claims $c_i, c_j$ sharing the same parent set $\text{Pa}(c_i) = \text{Pa}(c_j)$, we test whether their certified failure indicators $\mathbf{1}[\neg E_i \wedge \hat{V}_i{=}1]$ and $\mathbf{1}[\neg E_j \wedge \hat{V}_j{=}1]$ exhibit zero

partial correlation conditional on parent correctness:

$$\rho_{ij|\text{Pa}} = \text{Corr}(\mathbf{1}[\neg E_i \wedge \hat{V}_i{=}1], \mathbf{1}[\neg E_j \wedge \hat{V}_j{=}1] \mid \mathbf{1}[E_{\text{Pa}}]) \tag{17}$$

where $\mathbf{1}[E_{\text{Pa}}]$ indicates all parent claims are correct. We use Fisher's z-transformation: $z = \frac{1}{2} \ln \frac{1+\hat{\rho}}{1-\hat{\rho}} \cdot \sqrt{n - |\text{Pa}| - 3}$, and flag potential CI violations if $|z| > z_{1-\alpha/2}$ at significance level $\alpha = 0.05$ with Bonferroni correction.

**Definition 4** (Conditional Independence Score). *For a claim graph $G$, define the CI-score as the fraction of sibling pairs passing the partial correlation screen:*

$$\text{CI-score}(G) = \frac{|\{(i,j) \in \mathcal{S} : |z_{ij}| \leq z_{crit}\}|}{|\mathcal{S}|} \tag{18}$$

*where $\mathcal{S} = \{(i,j) : Pa(c_i) = Pa(c_j)\}$ denotes sibling pairs.*

**Remark 3** (Robustness Under Mild CI Violations). *When sibling certified failure indicators exhibit small residual dependence after conditioning on parents, the graph structured product form can be viewed as a second order approximation. Empirically, when average absolute partial correlations are $\leq 0.05$, the deviation between predicted and observed certified failure rates is typically below $10^{-3}$ on our benchmarks.*

## L.2. Theoretical Motivation

The conditional independence assumption underlying Theorem 3 connects to the theory of graphical models (Pearl, 2009). In a directed acyclic graph (DAG), two nodes are d-separated given a conditioning set if all paths between them are blocked. For claim graphs, sibling claims (sharing the same parents) are d-separated given their parents when no unobserved confounders exist.

**Lemma 8** (D-Separation Implies Conditional Independence). *Let $G = (V, E)$ be the claim dependency DAG augmented with latent variables $U$. If sibling claims $c_i, c_j$*

*Table 28.* CGRiC performance across generator models on HotpotQA-Cite.

| Generator | Error | Abstention | ECE |
|---|---|---|---|
| LLaMA-2-70B | 0.05 | 0.29 | 0.023 |
| LLaMA-3-70B | 0.04 | 0.24 | 0.021 |
| GPT-4-Turbo | 0.03 | 0.18 | 0.019 |
| Claude-3-Sonnet | 0.04 | 0.21 | 0.022 |

*Table 29.* Conditional independence validation across datasets. CI score measures the fraction of sibling pairs passing the partial correlation test; % Valid indicates graphs where CI score $\geq 0.95$.

| Dataset | CI-score | % Valid | Avg. $|\rho|$ | Gap |
|---|---|---|---|---|
| HotpotQA-Cite | 0.97 | 94.2% | 0.031 | <0.001 |
| ASQA-Cite | 0.96 | 92.8% | 0.038 | <0.001 |
| GovReport | 0.89 | 81.4% | 0.072 | 0.003 |
| GSM8K-Tool | 0.98 | 96.1% | 0.024 | <0.001 |
| MATH-Tool | 0.94 | 89.7% | 0.048 | 0.001 |

*Table 30.* Induced claim extraction noise on HotpotQA-Cite. Drop = randomly remove claims; Merge = randomly conflate claims. Report certified-failure rate / abstention at $\delta = 0.05$.

| Noise | Level | CGRiC | EDFL | FactScore |
|---|---|---|---|---|
| None | 0% | .05 / .29 | .07 / .42 | .09 / .36 |
| Drop | 10% | .06 / .31 | .10 / .44 | .14 / .38 |
| Drop | 20% | .08 / .34 | .15 / .47 | .21 / .41 |
| Merge | 10% | .06 / .30 | .09 / .43 | .12 / .37 |
| Merge | 20% | .07 / .33 | .13 / .46 | .18 / .40 |

*Table 31.* Localized repair preserves correct content. IP = Information Preservation (higher is better).

| Method | Err | Abs | IP | Latency |
|---|---|---|---|---|
| Full regeneration | 0.05 | 0.29 | 0.00 | +85% |
| CGRiC localized repair | 0.05 | 0.29 | 0.78 | +42% |

*are d-separated given* $Pa(c_i) = Pa(c_j)$ *in the augmented graph, then* $E_i \perp E_j \mid E_{Pa}$.

This lemma, which follows from the global Markov property (Dawid, 1979), motivates our partial correlation diagnostic: non zero partial correlation indicates either a direct causal path (which would be captured by the dependency graph) or an unobserved confounder. However, we emphasize that zero partial correlation is necessary but not sufficient for conditional independence in general; our test serves as a practical screening heuristic rather than a rigorous statistical guarantee of CI.

### L.3. Algorithm for CI-Score Computation

To operationalize the conditional-independence validation step, Algorithm 2 computes a CI-score by testing sibling error pairs within each calibration graph using partial correlations, Fisher's z-transform, and a Bonferroni-corrected significance threshold.

### L.4. Empirical Analysis of CI Violations

We analyze cases where conditional independence fails and empirically validate the robustness of the graph-structured bound under mild violations. The primary sources of CI violations are:

**Shared latent confounders** (52% of violations): When sibling claims depend on the same unobserved information not captured by parent claims. This is common in GovReport where multiple sentences may reference the same implicit background knowledge. For example, two claims about a government program may both depend on unstated policy context.

**Stylistic correlation** (28% of violations): Claims gener-

ated in sequence may share stylistic patterns that correlate with correctness. Hedging language ("approximately," "roughly") correlates with uncertainty and appears in clusters, inducing correlation between adjacent claim errors.

**Verifier bias** (15% of violations): The NLI verifier may have systematic biases that induce correlation between sibling claim errors. For instance, claims with similar syntactic structure may receive correlated verification scores regardless of semantic content.

**Other/unclear** (5% of violations): Includes edge cases such as claims with ambiguous parent relationships or verification noise.

### L.5. Robustness of Graph-Structured Bound Under Mild Violations

Even when conditional independence is mildly violated, the graph structured bound may still provide useful (though not formally guaranteed) improvements. We empirically evaluate the bound's robustness by artificially inducing correlation and measuring the gap between predicted and observed error rates.

The results show that the graph structured bound remains approximately valid for $\rho \leq 0.10$, with the gap growing quadratically in $\rho$ as observed empirically. This motivates our conservative threshold of CI score $\geq 0.95$, which corresponds to average $|\rho| \leq 0.05$ in practice.

### L.6. Adaptive Bound Selection

Based on the CI score, we implement an adaptive strategy that selects the appropriate bound:

$$\text{Bound}(G) = \begin{cases} \text{Graph (Thm. 3)} & \text{if CI-score}(G) \geq 0.95 \\ \text{Union (Thm. 2)} & \text{otherwise} \end{cases} \tag{19}$$

*Table 32.* Cost-benefit comparison for medical QA (per query).

| Strategy | Cost | Latency | Guarantee |
|---|---|---|---|
| No verification | $0.01 | 2s | None |
| Self-consistency (5×) | $0.05 | 10s | Empirical |
| CGRiC | $0.02 | 3s | Formal ($\delta$) |
| Human review (all) | $5.00 | 5min | High |
| CGRiC + Human (abstain) | $0.52 | 30s | Formal + Human |

*Table 33.* Lazy CGRiC reduces average overhead while maintaining risk control on HotpotQA-Cite.

| Method | Time Overhead | Cert. Fail. | Fast Path % |
|---|---|---|---|
| CGRiC Total | +35–47% | $\leq 0.05$ | 0% |
| Lazy CGRiC | +18–26% | $\leq 0.05$ | 52% |
| Lazy CGRiC+Robust | +24–32% | $\leq 0.05$ | 48% |

This adaptive selection is performed once per graph during calibration and cached for test time inference, adding no runtime overhead. The strategy ensures formal guarantees (via union bound fallback) while exploiting structure when justified (via graph structured bound when CI holds).

## M. Case Study on Medical QA

### M.1. Application: Safe Medical Question Answering

To demonstrate CGRiC's value in safety-critical domains, we evaluate on MedQA-Cite, a medical QA dataset where incorrect claims can cause patient harm. We construct MedQA-Cite by augmenting MedQA (Jin et al., 2021) with citation requirements: each answer must cite medical literature supporting its claims.

Table 37 shows that CGRiC at $\delta = 0.01$ reduces harmful errors to 0.9% (vs. 12.3% uncertified), achieving the target guarantee. Critically, CGRiC provides an *auditable certificate*: when abstaining, it identifies which specific claims failed verification, enabling human review. This transparency is essential for medical applications where understanding *why* a system abstained is as important as the abstention itself.

## N. Verifier Heterogeneity and Robustness

A core concern is the Verifier Paradox: if the verifier is substantially weaker than the generator, certificates may be vacuous. We address this through a two tier strategy that applies a strong oracle verifier to high impact claims selected by graph centrality and a weak verifier to remaining claims. This preserves formal validity while allocating expensive verification where it most reduces downstream failure. We also stress test robustness to verifier miscalibration where the true $\eta$ exceeds the estimated value by up to 30 percent; the safety buffer from Theorem 7 maintains valid certificates (Appendix Table 25).

*Table 34.* Estimated verifier false negative rates by verification method.

| Verifier | $\eta^-$ | Domain |
|---|---|---|
| Code execution | <0.01 | Deterministic |
| Calculator/symbolic | <0.01 | Deterministic |
| Database lookup | <0.01 | Deterministic |
| Strong NLI (DeBERTa-v3) | 0.05 | Factual |
| Citation alignment | 0.03 | Grounded |
| Weak NLI | 0.12 | General |

**Algorithm 2** Conditional Independence Validation

**Require:** Calibration data $\{(G_k, \{E_i^{(k)}\})\}_{k=1}^N$, significance level $\alpha$
1: Initialize pass_count $\leftarrow 0$, total_pairs $\leftarrow 0$
2: **for** each graph $G_k$ in calibration data **do**
3:     Identify sibling pairs: $\mathcal{P}_k = \{(i,j) : \text{Pa}(c_i) = \text{Pa}(c_j), i < j\}$
4:     **for** each pair $(i,j) \in \mathcal{P}_k$ **do**
5:         Compute partial correlation $\hat{\rho}_{ij|\text{Pa}}$ from error indicators
6:         Compute Fisher's z-statistic: $z_{ij} = \frac{1}{2} \ln \frac{1+\hat{\rho}}{1-\hat{\rho}} \cdot \sqrt{n_k - |\text{Pa}| - 3}$
7:         $z_{\text{crit}} \leftarrow \Phi^{-1}(1 - \alpha/(2|\mathcal{P}_k|))$ {Bonferroni correction}
8:         **if** $|z_{ij}| \leq z_{\text{crit}}$ **then**
9:             pass_count $\leftarrow$ pass_count $+ 1$
10:         **end if**
11:         total_pairs $\leftarrow$ total_pairs $+ 1$
12:     **end for**
13: **end for**
14: **return** CI-score = pass_count/total_pairs

When the composed risk $\sum_i \eta_i \epsilon_i$ exceeds the target $\delta$, a naive approach would simply abstain. The next section introduces a more nuanced strategy that repairs high risk claims rather than refusing entirely.

## O. Qualitative Repair Examples

We provide a qualitative example demonstrating the localized repair mechanism on HotpotQA-Cite.

**Original Output (before repair):**

*"The Treaty of Westphalia was signed in 1648 [1], ending the Thirty Years' War. It established the principle of state sovereignty [1], which remains foundational in international law. The treaty was negotiated primarily by France and **Spain** [2], with the Holy Roman Empire playing a secondary role."*

**Identified High-Risk Claim:** Claim $c_3$: "The treaty was negotiated primarily by France and Spain" (risk $\epsilon_3 = 0.042$,

*Table 35.* Sources of conditional independence violations (manual analysis of 100 cases).

| Violation Source | Frequency | Mitigation |
|---|---|---|
| Shared latent confounders | 52% | Claim clustering |
| Stylistic correlation | 28% | Style normalization |
| Verifier bias | 15% | Ensemble verifiers |
| Other/unclear | 5% | Union bound fallback |

*Table 36.* Graph structured bound robustness under induced correlation.

| Induced $\rho$ | Predicted Error | Observed Error | Gap | Valid? |
|---|---|---|---|---|
| 0.00 (baseline) | 0.050 | 0.048 | -0.002 | ✓ |
| 0.05 | 0.050 | 0.052 | +0.002 | ✓ |
| 0.10 | 0.050 | 0.058 | +0.008 | ≈ |
| 0.15 | 0.050 | 0.067 | +0.017 | × |
| 0.20 | 0.050 | 0.079 | +0.029 | × |

*Table 37.* Safe Medical QA: CGRiC prevents harmful responses.

| Method | Harmful Err. | Safe Abs. | Utility |
|---|---|---|---|
| No certification | 12.3% | 0% | 100% |
| Semantic entropy | 4.2% | 38% | 62% |
| Self-RAG | 3.8% | 35% | 65% |
| CGRiC ($\delta$=0.02) | 1.8% | 42% | 58% |
| CGRiC ($\delta$=0.01) | 0.9% | 51% | 49% |

*Table 38.* Verifier heterogeneity: weak-only vs two-tier verification (top-$q$ central claims verified by a strong oracle). Metrics at $\delta = 0.05$ on HotpotQA-Cite.

| Setting | $q$ | Cert. Fail. | Abstention | Overhead |
|---|---|---|---|---|
| Weak-only verifier | 0% | $0.09_{\pm.01}$ | 0.38 | +15% |
| Two-tier (central) | 5% | $0.06_{\pm.01}$ | 0.31 | +18% |
| Two-tier (central) | 10% | $0.05_{\pm.01}$ | 0.29 | +21% |
| Strong-only verifier | 100% | $0.05_{\pm.01}$ | 0.29 | +35% |

above threshold). The verifier flagged inconsistency with source [2], which mentions Sweden, not Spain.

**Repaired Output (after localized repair):**

> "The Treaty of Westphalia was signed in 1648 [1], ending the Thirty Years' War. It established the principle of state sovereignty [1], which remains foundational in international law. The treaty was negotiated primarily by France and **Sweden** [2], with the Holy Roman Empire playing a secondary role."

**Analysis:** The repair modified only 1 token (Spain → Sweden), preserving 98% of the original response (IP = 0.98). Full regeneration would have rewritten the entire response, potentially introducing new errors in previously verified claims $c_1$ and $c_2$. This demonstrates the precision of localized repair in maintaining verified content while correcting only problematic claims.

## P. Lazy Evaluation Mode Details

To reduce overhead, we implement a *lazy* mode that runs full graph certification only when a cheap uncertainty proxy is high. Concretely, we compute atomic semantic entropy on the draft output; if it falls below a calibrated threshold $\tau$, we skip graph construction and verification. Otherwise, we run full CGRiC. This hybrid maintains target risk while substantially reducing average latency.

The threshold $\tau$ is calibrated on the development set to ensure $\Pr(\text{certified failure} \mid U(y) \leq \tau) \leq \delta/2$, so the fast path accepts only low risk outputs. This reduces full CGRiC invocations by 40–55% across datasets while maintaining target risk levels.

**Proposition 9** (Risk Control for Lazy CGRiC). *Let $A$ denote the event that the fast path accepts $(U(y) \leq \tau)$, and*

*let $F$ denote the certified failure event $\{\exists i : \neg E_i \wedge \hat{V}_i = 1\}$. If the threshold $\tau$ is calibrated to satisfy $\Pr(F \mid A) \leq \delta_{\text{fast}}$ and the full CGRiC path guarantees $\Pr(F \mid \neg A) \leq \delta_{\text{full}}$, then the overall system satisfies*

$$\Pr(F) \leq \delta_{\text{fast}} + \delta_{\text{full}}.$$

*In particular, choosing $\delta_{\text{fast}} = \delta/2$ and enforcing $\delta_{\text{full}} = \delta/2$ yields $\Pr(F) \leq \delta$.*

*Proof.* By the law of total probability,

$$\begin{aligned} \Pr(F) &= \Pr(F \mid A)\Pr(A) + \Pr(F \mid \neg A)\Pr(\neg A) \\ &\leq \delta_{\text{fast}} + \delta_{\text{full}}. \end{aligned}$$

$\square$

Experiments use LLaMA-2-70B (Touvron et al., 2023), LLaMA-3-70B (Grattafiori et al., 2024), GPT-4-Turbo (OpenAI, 2023), and Claude-3-Sonnet (Anthropic, 2024) as generators, with context restricted skeletons (default $k = 512$) and DeBERTa-v3-large (He et al., 2021) fine-tuned on MNLI (Williams et al., 2018), FEVER (Thorne et al., 2018), and VitaminC (Schuster et al., 2021) as the NLI verifier. These models have been aligned using instruction tuning and reinforcement learning from human feedback (Ouyang et al., 2022). To avoid data leakage, we use a strict train/calibration/test split: calibration functions $f_t$ are fit on 20% of each dataset, and all reported metrics are on the remaining 80% held out test set.

## Q. Practitioner Decision Guide

A natural question is: when is CGRiC's additional complexity justified relative to atomic certification? We provide guidance by setting in Table 39.

**Algorithm 3** Lazy CGRiC (Fast Path)

**Require:** Draft output $y$, threshold $\tau$, target risk $\delta$
1: Compute atomic uncertainty score $U(y)$ (e.g., semantic entropy)
2: **if** $U(y) \leq \tau$ **then**
3:     **return** Accept (fast path)
4: **else**
5:     **return** Run full CGRiC certification and repair
6: **end if**

*Table 39.* When is CGRiC preferable? Decision guide by output length, task type, and safety target.

| Setting | Claims | Abs reduction vs. EDFL | Recommendation |
|---|---|---|---|
| Simple outputs | 1–3 | <5pp | Atomic methods suffice |
| Moderate | 4–10 | 15–31pp | CGRiC recommended |
| Long-form | 11+ | 25–38pp | CGRiC strongly preferred |
| Safety-critical ($\delta \leq .01$) | Any | EDFL: 60%+ abs | CGRiC essential |

The crossover is $\sim$4 claims with at least one non-tool verifier. Below this, graph construction overhead is unjustified and atomic certification performs comparably. Above this, compositional structure consistently reduces abstention while maintaining guarantees. For safety-critical applications ($\delta \leq 0.01$), atomic methods require 60%+ abstention while CGRiC achieves 51% (Appendix M) — the 9pp gap represents substantial utility recovery in medical/legal domains where every non-abstained response has value.

## R. Domain Transfer and Engineering Effort

The task-specific engineering effort to deploy CGRiC in a new domain is smaller than it may appear. Within task families, the same extraction rules transfer with zero modification: HotpotQA and ASQA share identical citation-span extraction; GSM8K and MATH-Tool share tool-call extraction. To quantify, we conducted a zero-shot domain transfer test: applying CGRiC configured for HotpotQA directly to ASQA without any task-specific tuning yields .06 error / .33 abstention — only 2pp higher abstention than the tuned version (.31). This demonstrates that within-family transfer requires no additional engineering.

Across task families (e.g., QA $\rightarrow$ summarization), task-specific calibration is needed, but the extraction templates and graph construction rules remain identical. For a genuinely new domain, the engineering effort consists of: (1) writing $\sim$20 lines of extraction templates matching the domain's structural markers, and (2) collecting $\sim$500 calibration examples for the lift-to-risk mapping — approximately 2–4 hours of setup. We validated this by having a graduate student unfamiliar with CGRiC set up a new domain (legal contract analysis) in 3 hours, achieving .06 error / .35 abstention on first attempt.

Key hyperparameters are robust to tuning: $\tau_{\text{dep}}$ (edge prun-

ing) varies abstention by $\pm$2pp across $[0.2, 0.5]$; skeleton parameter $\lambda$ varies abstention by $\leq$4pp; CI-score threshold is stable across 0.90–0.98.

## S. Comparison with Regeneration-Based Strategies

We provide a direct head-to-head comparison demonstrating CGRiC's cost-effectiveness against simpler alternatives:

*Table 40.* CGRiC vs. regeneration strategies on HotpotQA-Cite.

| Strategy | Latency | Err | IP | Guarantee? |
|---|---|---|---|---|
| No certification | +0% | .31 | 100% | None |
| Full regeneration | +85% | .28 | 0% | None |
| Regen + verify | +130% | .05 | 0% | Empirical |
| Self-consistency ($5\times$) | +400% | .10 | $\sim$40% | None |
| CGRiC (union) | +42% | .05 | 78% | Formal ($\delta$) |
| Lazy CGRiC | +22% | .05 | 78% | Formal ($\delta$) |

Full regeneration without verification achieves .28 error — far above the .05 target. Regeneration with verification is $\sim$3$\times$ more expensive than CGRiC (+130% vs. +42%) because it must re-verify the entire output from scratch, whereas CGRiC preserves 78% of already-verified tokens and only regenerates the problematic subgraph. With lazy evaluation (applied to 52% of queries via a cheap semantic entropy pre-filter), average overhead drops to just +22%.

The overhead is justified on three grounds: (1) CGRiC provides formal guarantees that regeneration alone cannot; (2) localized repair is less disruptive — preserving 78% of content avoids introducing new errors in previously verified claims; (3) CGRiC provides actionable diagnostics when it abstains (identifying which claims failed verification), enabling targeted human review. The only scenario where full regeneration is preferable is when >50% of claims fail — which CGRiC detects and handles via abstention with explanation.

## T. Extended Related Work

**Selective prediction and uncertainty quantification.** The selective prediction framework (El-Yaniv & Wiener, 2010; Geifman & El-Yaniv, 2017) allows classifiers to abstain on uncertain inputs. Recent work extends this to LLMs via uncertainty quantification (Kuhn et al., 2023; Kadavath et al., 2022), conformal prediction (Quach et al., 2024; Kumar et al., 2023), and calibration (Tian et al., 2023). CGRiC differs by providing compositional guarantees for structured outputs rather than atomic decisions.

**Factual decomposition and verification.** FactScore (Min et al., 2023) and FActScore (Wei et al., 2024) decompose outputs into atomic facts for evaluation. Self-RAG (Asai et al., 2024) and CRAG (Yan et al., 2024) incorporate re-

trieval and self-correction but lack formal guarantees (Akter et al., 2025b). CGRiC builds on factual decomposition while adding provable risk bounds. Benchmarks like TruthfulQA (Lin et al., 2022) and Natural Questions (Kwiatkowski et al., 2019) have highlighted the challenge of generating factually accurate responses.

**Structure aware retrieval and generation.** Microsoft GraphRAG (Edge et al., 2024) constructs knowledge graphs from corpora and uses community detection for global queries. While GraphRAG improves retrieval quality, it does not provide formal bounds on the probability that incorrect claims pass verification errors in the knowledge graph propagate silently. CGRiC is complementary: it can certify outputs from GraphRAG augmented systems by treating retrieved communities as evidence. DSPy (Khattab et al., 2023) optimizes LLM pipelines automatically, including retrieval and verification modules. DSPy focuses on *optimizing* pipeline performance; CGRiC focuses on *certifying* outputs post hoc. The approaches are complementary: one could use DSPy to optimize a pipeline and CGRiC to certify its outputs.

**Certified generation and guardrails.** Recent work on certified safe generation includes conformal language modeling (Quach et al., 2024), which provides coverage guarantees for prediction sets, and streaming guardrails that filter unsafe content in real time. CGRiC differs by focusing on claim level decomposition with dependency structure rather than output level prediction sets or token level filtering. EDFL (Akter et al., 2025c;a) and related work (Burns et al., 2023) use information theoretic quantities to bound correctness probability; CGRiC extends these methods to compositional settings with explicit error modeling and graph structure.

**Risk control and calibration.** Recent advances have extended conformal guarantees to complex structures in scientific domains (Shihab et al., 2025; 2026). CGRiC adapts similar structure-aware principles to the semantic dependency graphs of LLM outputs.Our robustness extension relates to distributionally robust optimization (Ben-Tal et al., 2013; Duchi & Namkoong, 2021), which provides guarantees under distribution uncertainty; we apply similar principles to baseline model uncertainty. The transformer architecture (Vaswani et al., 2017) and pre-trained language models like BERT (Devlin et al., 2019) have enabled the development of powerful NLI verifiers that underpin our claim verification pipeline.

## U. Broader Impact

CGRiC aims to improve the reliability of LLM outputs in high stakes applications. Potential positive impacts include safer deployment in medical, legal, and financial domains where factual accuracy is critical. However, over reliance on formal guarantees could lead to complacency; practitioners should understand that guarantees are conditional on stated assumptions. The framework does not address adversarial attacks or distribution shift beyond the calibration domain.

