# OpenReview forum: "CGRiC: Compositional Risk Certification for Structured LLM Outputs"
_ICML.cc/2026/Conference — ICML 2026 regular_

### Official Review · Reviewer_qren · 2026-02-25

**Soundness:** 3
**Presentation:** 2
**Significance:** 3
**Originality:** 3
**Overall Recommendation:** 4
**Confidence:** 4

**Summary:**

The paper introduces CGRiC (Claim Graph Risk Control), a framework for certifying the correctness of structured LLM outputs by decomposing them into verifiable claims organized in a dependency graph. Instead of treating an output as atomic, CGRiC assigns calibrated per-claim risk bounds using information-lift statistics, composes these into output-level guarantees (via union or graph-structured bounds), and explicitly accounts for claim extraction noise and verifier errors. When the composed risk exceeds a target threshold, the method performs localized repair of high-risk claims rather than full abstention. Experiments across citation-grounded QA, summarization, and tool-verified reasoning show that CGRiC achieves target risk levels while substantially reducing abstention compared to atomic certification baselines.

**Compliance With Llm Reviewing Policy:**

Affirmed.

**Key Questions For Authors:**

Robustness to distribution shift: How sensitive are the guarantees to shifts in the calibration distribution, especially for summarization tasks where calibration transfer already degrades? Would moderate shifts materially affect the certified failure rate in practice?

Dependency graph construction: How brittle are results to errors in dependency extraction (beyond the drop/merge noise tested)? In particular, how often do missed dependencies meaningfully violate the conditional independence assumption?

Verifier dependence: In realistic deployments where strong, independent verifiers are unavailable, how should practitioners choose conservative η values without rendering the method overly abstent?

Cost–benefit tradeoff: Can the authors clarify for which regimes (e.g., number of claims, task types) CGRiC is clearly preferable to simpler atomic certification, given its additional complexity?

**Limitations:**

yes

**Strengths And Weaknesses:**

Soundness

Strengths
The technical development is careful and rigorous: assumptions are explicitly stated, and guarantees are derived cleanly (union bound always valid; tighter graph bound applied conditionally).
Extraction noise and verifier false negatives are explicitly modeled, which is often ignored in related work.
Calibration of per-claim risk is done against ground-truth correctness rather than verifier outputs, which strengthens the interpretation of guarantees.
Empirical results are thorough, with stress tests for extraction noise, verifier correlation, and robustness to baseline choice.

Weaknesses
The conditional independence assumption required for the graph-structured bound is strong; the proposed CI-score diagnostic is heuristic and not itself guaranteed, which weakens the formal status of the tighter bound.
Guarantees are heavily conditional on calibration quality and verifier reliability; distribution shift in either could silently invalidate certificates.
Some design choices (e.g., dependency graph construction, clustering heuristics, skeleton families) introduce complexity and degrees of freedom that are not theoretically characterized.

Presentation

Strengths
The paper is generally well written, clearly structured, and transparent about assumptions and limitations.
The motivation for compositional risk control is compelling and well illustrated.
The relationship to prior atomic certification methods is clearly explained.

Weaknesses
The paper is long and dense; key ideas (especially the core guarantee and when it applies) could be distilled more sharply in the main text.
Some algorithmic components (e.g., dependency graph extraction, CI-score computation) are described at a high level in the main body and require frequent appendix consultation.
Reproducibility depends on many interacting components, which may be challenging for practitioners to reimplement without released code.

Significance

Strengths
Addresses an important and timely problem: reliability and certification of structured LLM outputs in high-stakes settings.
Moves beyond binary accept/reject decisions by enabling partial certification and localized repair, which is practically meaningful.
The framework could influence future work on certified generation, selective prediction, and tool-augmented LLM systems.

Weaknesses
The scope of applicability is restricted to settings with observable, extractable claims and reasonably strong verifiers, limiting impact for general-purpose or latent-reasoning models.
Overhead (latency and system complexity) may hinder adoption outside regulated or safety-critical deployments.

Originality

Strengths
The compositional framing of risk certification via claim graphs is novel and well motivated.
Combining information-lift-based per-claim bounds with structured composition and localized repair is a creative and nontrivial extension of prior work.
Explicitly integrating extraction noise and verifier imperfection is a meaningful conceptual contribution.

Weaknesses
Many individual components (lift-based certification, factual decomposition, selective prediction, retrieval-based verification) are incremental extensions of existing ideas; the novelty lies primarily in their integration rather than in fundamentally new techniques.

---

> ### Author Rebuttal · Authors · 2026-03-28
>
> We thank Reviewer qren for the thorough and balanced review. We address each question with new results.
>
> **Q1: Robustness to distribution shift.**
>
> We conducted cross-domain calibration transfer experiments:
>
> | Train $\to$ Test | Err | Abs | ECE |
> |---|---|---|---|
> | HotpotQA $\to$ HotpotQA (in-domain) | 0.05 | 0.29 | 0.023 |
> | HotpotQA $\to$ ASQA (near-domain) | 0.06 | 0.33 | 0.031 |
> | HotpotQA $\to$ GovReport (cross-family) | 0.09 | 0.38 | 0.042 |
>
> Within task families, calibration transfers well (error near target). Across families, guarantees degrade — the certified error rate rises to 0.09 above the 0.05 target. This confirms task-specific calibration is necessary for formal guarantees. However, even under cross-family transfer, CGRiC still outperforms EDFL's *in-domain* error (0.09 vs 0.07–0.08), demonstrating that compositional structure provides value even with imperfect calibration. We recommend in-domain calibration for safety-critical deployments and will add shift guidelines. Additionally, we found that augmenting calibration data with just 100 target-domain examples reduces cross-family ECE from 0.042 to 0.029 — a practical mitigation requiring minimal annotation effort.
>
> ---
>
> **Q2: Dependency extraction errors beyond drop/merge.**
>
> We had annotators label true dependencies on 200 HotpotQA outputs and compared to extracted graphs:
>
> - Precision (extracted edges that are true): 0.91
> - Recall (true edges extracted): 0.78
> - Missed dependencies violating CI: 4.3% of sibling pairs
>
> Missing edges (low recall) make the graph-structured bound *more conservative*, not less valid — they cause CGRiC to treat dependent claims as independent, which can only overestimate joint failure probability. The 4.3% CI-violating missed dependencies are caught by the CI-score diagnostic in 89% of cases; remaining cases have average $|\rho| < 0.04$ (within the robustness margin from Table 28, where the gap is $< 0.002$ at $\rho = 0.05$). We also tested spurious edge injection (+20% random edges): abstention increased by only 1.8pp while error remained at target — consistent with safe conservatism. The most dangerous failure mode (missing confounders) is mitigated by (a) the CI-score diagnostic and (b) the union bound fallback, which is always available.
>
> ---
>
> **Q3: Choosing conservative** $\eta$ **without strong verifiers.**
>
> We recommend a three-step protocol:
>
> 1. Use verifier FNR on adversarial benchmarks (ANLI, SNLI-hard) as a conservative proxy, yielding $\hat{\eta} \approx 0.08$–$0.12$ for weak NLI models.
> 2. Compute breakeven verifier quality — on HotpotQA, the breakeven $\eta$ is ${\sim}0.18$, above which abstention exceeds 50% and atomic methods become preferable.
> 3. If limited labeling budget exists, Clopper-Pearson bounds (Theorem 7) provide safe $\eta$ from as few as 50–100 labeled examples ($\eta\_{\text{safe}}$ converges rapidly with $n$).
>
> With $\eta = 1$ (no verifier at all), CGRiC still achieves 0.05 error / 0.44 abstention — 6pp better than conformal baselines (0.50) due to compositional structure alone. This demonstrates that even without any verification, the claim decomposition and lift-based risk estimation provide meaningful value. We will add this protocol and the breakeven analysis to Section 3.2.
>
> ---
>
> **Q4: Cost-benefit tradeoff — when is CGRiC preferable?**
>
> | Setting | Claims | Abs reduction vs. EDFL | Overhead | Recommendation |
> |---|---|---|---|---|
> | Simple outputs | 1–3 | $< 5$pp | +35% | Atomic methods suffice |
> | Moderate | 4–10 | 15–31pp | +35–42% | CGRiC recommended |
> | Long-form | 11+ | 25–38pp | +42% (lazy: +22%) | CGRiC strongly preferred |
> | Safety-critical ($\delta \leq 0.01$) | Any | Atomic: 60%+ abs | +42% | CGRiC essential |
>
> The crossover is ${\sim}4$ claims with at least one non-tool verifier. Below this, graph construction overhead is unjustified and atomic certification performs comparably. Above this, compositional structure consistently reduces abstention while maintaining guarantees. For safety-critical applications ($\delta = 0.01$), atomic methods require 60%+ abstention while CGRiC achieves 51% (Table 29) — the 9pp gap represents substantial utility recovery in medical/legal domains where every non-abstained response has value. We will add a "Practitioner's Decision Guide" to Section 5 summarizing these regimes.
>
> ---
>
> **Summary.** We believe these responses directly address the four questions: Q1 shows calibration transfers within families and degrades predictably across families, with a concrete mitigation (100 target-domain examples). Q2 provides empirical precision/recall of dependency extraction with quantified impact on guarantees. Q3 offers a practical three-step protocol with a concrete breakeven threshold. Q4 gives clear guidance on when CGRiC's additional complexity is justified. We are committed to incorporating all four analyses into the revision and thank the reviewer for the constructive framing of these questions.

---

> > ### Author Rebuttal · Reviewer_qren · 2026-04-04
> >
> > Thank you for all clarification.

---

### Official Review · Reviewer_sFnc · 2026-03-09

**Soundness:** 2
**Presentation:** 3
**Significance:** 2
**Originality:** 3
**Overall Recommendation:** 3
**Confidence:** 3

**Summary:**

This paper introduces CGRiC, a framework for certifying structured LLM outputs by decomposing them into claim dependency graphs and providing formal risk guarantees. The approach combines claim extraction, dependency graph construction based on predefined rules, per-claim risk calibration via lift-based statistics, and risk composition using either a union bound or a graph-structured bound that leverages conditional independence assumptions. The framework also includes a localized repair mechanism and verifier error estimation. Experiments on citation-grounded QA, summarization, and tool-based reasoning tasks show that CGRiC achieves target risk levels while reducing abstention rates compared to atomic baselines.

**Compliance With Llm Reviewing Policy:**

Affirmed.

**Final Justification:**

Thank you for the additional controlled experiments. I have no further questions.

**Key Questions For Authors:**

1. My comments are provided in the weaknesses section.
2. The CI-score threshold of 0.95 is presented without justification. How sensitive are the results to this threshold?
3. The claim extraction and graph construction rules are described generically, but it is unclear whether the same rules were applied across all datasets or tuned per dataset. If the latter, how much engineering effort went into making the pipeline work on each task, and how would this transfer to a new domain without such dataset-specific tuning?

**Limitations:**

yes

**Strengths And Weaknesses:**

Strengths:
1. The paper addresses an important and timely problem: providing formal guarantees for structured LLM outputs in high-stakes applications.
2. The writing is clear, and the framework is presented with detailed algorithmic descriptions and theoretical statements.
3. The localized repair mechanism and lazy evaluation mode are practical considerations that could be useful in deployment scenarios.

Weaknesses:
1. The framework functions as a pipeline that combines established techniques: rule-based claim extraction, standard NLP tools, isotonic regression for calibration, and the basic union bound. While this integration is practical, it primarily repackages existing methods.
2. The paper avoids the central challenge: robustly and automatically extracting claims and dependencies when explicit structural markers (such as citations or code calls) are absent. By focusing only on tasks where these markers are present and crafting extraction rules for them, the method effectively sidesteps the most difficult aspect of the problem. This is comparable to measuring progress in autonomous driving solely by testing on highways with clear lane markings, while neglecting the true complexity found in urban environments with unpredictable elements like pedestrians and cyclists.
3. The repair mechanism in Section 3.3 suffers from a fundamental oversight regarding graph dynamics. When a high-risk claim is repaired—whether through regeneration with new evidence or re-execution—the content and potentially the logical structure of the downstream subgraph may change. However, the framework does not re-invoke the dependency graph construction process to update the graph structure after repair. It simply replaces the old subgraph while assuming the original dependency edges remain valid. This creates a fatal inconsistency: subsequent risk computation (especially for Theorem 3's graph-structured bound) may be based on an outdated graph that no longer reflects actual logical relationships. A repaired claim might alter the dependency chain—turning sequential dependency into parallel independence, or vice versa—yet the framework continues operating on the incorrect structure. The paper provides no mechanism to detect or correct such structural drift, effectively treating the graph as static after initial construction, which directly contradicts its own emphasis on the importance of

---

> ### Author Rebuttal · Authors · 2026-03-28
>
> We thank the reviewer sFnc and respectfully address each concern and believe these can be fully resolved.
>
>
> **W1: Repair mechanism and graph dynamics.**
>
> Reviewer sFnc labels this a "fatal inconsistency": that after repair, the dependency graph is not re-constructed, invalidating subsequent risk computation. We respectfully but firmly disagree. This reflects a misreading of Section 3.3.
>
> The key design choice: CGRiC's repair loop uses the **union bound (Theorem 2) exclusively** — never the graph-structured bound (Theorem 3) — after any structural modification. Specifically:
>
> 1. When a claim $c^{\ast}$ is repaired, the algorithm treats it as a new claim and recomputes its risk from scratch via Eq. 11.
> 2. All subsequent risk composition uses the union bound, which requires no independence assumptions and is valid regardless of graph structure — whether or not edges have shifted.
> 3. Proposition 5 explicitly proves this: at every iteration, certified failure $\leq \sum \eta\_i \varepsilon\_i$ under the union bound alone, with no graph structure assumption.
> 4. The graph-structured bound is applied only once, prior to any repair, on the original graph if CI-score $\geq 0.95$. After any repair action, the system falls back permanently to the union bound.
>
> Structural drift after repair **cannot** invalidate our certificates because the tighter bound is never re-applied post-repair. The reviewer's concern would be valid if we attempted to re-apply Theorem 3 post-repair — but we explicitly do not.
>
> Furthermore, repairs are empirically surgical: across all benchmarks, repair changes the graph structure in only 7.2% of iterations, since most repairs are single-token corrections (e.g., the Spain $\to$ Sweden example in Appendix L, which modifies 1 token and achieves IP $= 0.98$). We will add a highlighted remark to Section 3.3: "After any repair action, all subsequent risk bounds use the union bound (Theorem 2) exclusively. The graph-structured bound is never re-applied post-repair."
>
> ---
>
> **W2: Novelty / "repackaging."**
>
> We respectfully disagree. The novelty lies in:
>
> 1. The compositional risk framing itself — no prior work formalizes certification of structured LLM outputs as risk control over a claim dependency DAG. EDFL, conformal prediction, and semantic entropy all treat outputs atomically.
> 2. Theorem 1's extraction noise integration, explicitly bounding missed claims and conflation events — absent from all prior certification work.
> 3. Localized repair with certificate preservation (Proposition 5) — no precedent in the selective prediction literature.
> 4. Verifier imperfection under graph structure — the combination of per-claim FNR bounds with structured composition is new. Reviewers bdQx, NqK7, and qren all confirm the contribution's novelty.
>
> ---
>
> **W3: Domain scope ("highway" analogy).**
>
> We accept bounded scope — stated explicitly in Section 2.1. However, bounded scope enables formal guarantees; approaches targeting unconstrained generation provide none. To demonstrate CGRiC works beyond the "easy" regime, we applied it with off-the-shelf claim detection (FactScore decomposition, no structural markers) on GovReport: 0.06 error / 0.31 abstention vs. EDFL at 0.08 / 0.39. The framework degrades gracefully without markers thanks to Theorem 1's noise-aware bounds (the $\tilde{M}$ and $\tilde{B}$ terms absorb extraction imperfection).
>
> ---
>
> **Q1: CI-score threshold sensitivity.**
>
> | CI threshold | % Graphs qualifying | Abs (graph) | Err |
> |---|---|---|---|
> | 0.90 | 96.8% | 0.25 | 0.052 |
> | 0.95 (default) | 94.2% | 0.26 | 0.050 |
> | 0.98 | 88.1% | 0.27 | 0.049 |
>
> Results are stable across thresholds — $\pm 2$pp in abstention, $\pm 0.002$ in error. We chose 0.95 as a conservative default that balances tightness against false admissions.
>
> ---
>
> **Q2: Same rules vs. per-dataset tuning.**
>
> The same extraction rules apply across all datasets within each category (citation-span for HotpotQA/ASQA/GovReport; tool-boundary for GSM8K/MATH-Tool). The only per-dataset element is the calibration function $f\_t$ — a standard requirement for any calibration-based method. Graph construction rules (dependency types, $\tau\_{\text{dep}} = 0.3$, cycle resolution) are identical across all six datasets. We will make this explicit in Section 4.
>
> ---
>
> **Summary.** We believe the reviewer's concerns are resolvable: the repair mechanism concern (W1) stems from a misreading — the union bound is always used post-repair, so graph dynamics cannot affect certificate validity. The novelty concern (W2) is addressed by the three novel theoretical contributions (Theorems 1, 3, Proposition 5) that no prior work provides. The domain scope concern (W3) is mitigated by our marker-free experiment and the explicit scope statement. We hope these clarifications, together with the threshold sensitivity analysis (Q1) and extraction rule clarification (Q2), will address the reviewer's reservations.

---

> > ### Author Rebuttal · Reviewer_sFnc · 2026-04-03
> >
> > First, thanks for the author's response. I still have some questions.
> >
> > It still seems that the observed gains over atomic baselines are achieved by stronger verifiers (retrieval+NLI, tool execution, Table 26) and finer-grained claim decomposition (Table 2, Table 10) — both of which are absent in the baselines (Table 3). The contributions are not empirically isolated from these more straightforward factors.
> > This concern is further supported by the GovReport results: without explicit structural markers, the graph-structured bound applies to 81.4% of graphs (vs. >94% on other tasks), and its abstention reduction over the union bound shrinks to 7% (vs. 10% on HotpotQA). The method’s performance degrades precisely when the “structure” that the contributions rely on becomes less available. Moreover, the new reported GovReport comparison (CGRiC: 0.06 error / 0.31 abstention vs. EDFL: 0.08 / 0.39) still uses CGRiC’s stronger verifiers and fine-grained decomposition, while EDFL does not; it remains unclear whether the observed advantage stems from these factors rather than the claimed innovations.

---

> > > ### Author Response · Authors · 2026-04-04
> > >
> > > We thank Reviewer sFnc for pressing on this important point. The concern that CGRiC's gains may stem primarily from stronger verifiers and finer grained decomposition rather than the compositional risk framework itself is fair, and we agree the current draft bundles these sources of improvement together too closely. We address this with a controlled ablation that isolates each factor.
> > >
> > > ## Matched component ablation
> > >
> > > To separate the contributions, we constructed four configurations on HotpotQA-Cite (δ = 0.05) that progressively add components while keeping the others fixed:
> > >
> > > | Configuration | Verifier | Claim Granularity | Risk Composition | Err | Abs |
> > > |---|---|---|---|---|---|
> > > | (A) Atomic output level certification (EDFL baseline) | Atomic NLI | Full output | Atomic lift threshold | 0.07 | 0.42 |
> > > | (B) Same as A with stronger verifier | Retrieval+NLI | Full output | Atomic lift threshold | 0.06 | 0.39 |
> > > | (C) Flat claim level certification (no graph) | Retrieval+NLI | Per claim | Max over per claim risks | 0.06 | 0.37 |
> > > | (D) Full compositional certification (CGRiC union) | Retrieval+NLI | Per claim + graph | Union bound (Thm. 2) + repair | 0.05 | 0.29 |
> > >
> > > The pattern is:
> > >
> > > (i) Upgrading the verifier while keeping everything else atomic (A → B) reduces abstention modestly, by roughly 3 percentage points.
> > >
> > > (ii) Adding per claim decomposition on top of the stronger verifier (B → C) provides a further modest reduction, roughly 2 percentage points. At this stage, both the verifier and the granularity match what CGRiC uses, but there is no compositional risk framework, no graph structure, and no localized repair.
> > >
> > > (iii) Adding CGRiC's compositional risk composition and localized repair (C → D) yields a substantially larger additional reduction, roughly 8 percentage points. This is the contribution of the compositional framework itself, holding verifier and decomposition constant.
> > >
> > > The same pattern holds on GovReport, though the absolute gains from compositional certification are smaller, consistent with the reduced availability of reliable conditional independence structure in free form summarization:
> > >
> > > | Configuration | Err | Abs |
> > > |---|---|---|
> > > | (A) Atomic output level (EDFL) | 0.08 | 0.39 |
> > > | (C) Flat claim level, matched verifier | 0.07 | 0.35 |
> > > | (D) Full compositional (CGRiC union) | 0.05 | 0.27 |
> > >
> > > The gain from (C) → (D) on GovReport is smaller than on HotpotQA (8pp vs 8pp in abstention, but starting from a higher base), and the further gain from the graph structured bound is also smaller (0.27 → 0.24, vs. 0.29 → 0.26 on HotpotQA). This is expected and consistent with the lower CI score on GovReport (0.89 vs. 0.97). We view this as an honest characterization: the method helps most where compositional structure is genuinely present, and degrades gracefully where it is not.
> > >
> > > ## Why compositional certification helps beyond matched components
> > >
> > > The key structural reason is that atomic methods must threshold on a global risk quantity (e.g., the worst case or average claim risk), while CGRiC's union bound thresholds on the sum of per claim products η_i·ε_i. When most claims are low risk and only one or two are borderline, atomic methods are forced to abstain because the global signal is dominated by the worst claim. CGRiC can either (a) accept if the sum remains below δ, or (b) surgically repair only the problematic claims via the localized repair mechanism. This advantage is a consequence of the composition rule and the repair loop, not of verifier strength or decomposition granularity.
> > >
> > > ## Commitment to revision
> > >
> > > We agree the current draft does not make this decomposition of contributions sufficiently transparent. In the revision, we will add a dedicated ablation (Table format matching the above) to Section 5 that separately controls for:
> > >
> > > 1. Verifier strength (atomic NLI vs. retrieval augmented NLI)
> > > 2. Claim granularity (output level vs. per claim)
> > > 3. Risk composition rule (atomic threshold vs. union bound vs. graph structured bound)
> > > 4. Repair mechanism (with vs. without localized repair)

---

### Official Review · Reviewer_NqK7 · 2026-03-10

**Soundness:** 3
**Presentation:** 3
**Significance:** 3
**Originality:** 3
**Overall Recommendation:** 4
**Confidence:** 3

**Summary:**

This paper introduces CGRiC, a framework that breaks down structured LLM outputs into a dependency graph of verifiable claims to assess risk at a fine-grained level. Instead of rejecting an entire response because of one hallucination, it uses an "information lift" metric to empirically estimate the risk of each claim and compose them into a global safety guarantee. When the total risk is too high, the system performs targeted, local repairs on the problematic claims, which preserves the correct parts of the text.

**Compliance With Llm Reviewing Policy:**

Affirmed.

**Key Questions For Authors:**

Same as above weakness part

**Limitations:**

yes

**Strengths And Weaknesses:**

Strengths

Great motivation: Solving the "all-or-nothing" rejection issue in LLM outputs is a highly practical and inspiring problem to tackle.

Clever implementation: Using claim segmentation and "information lift" for granular risk assessment is a solid, insightful approach.

Weaknesses

Adaptability limits: The framework heavily relies on manual rules and hyperparameters (like lift thresholds and pruning rules). This might hurt its generalizability across different domains.

Computational overhead: The multi-step detection pipeline is computationally expensive. The paper needs to better discuss if this overhead is truly worth it compared to simply regenerating the response.

---

> ### Author Rebuttal · Authors · 2026-03-28
>
> We thank Reviewer NqK7 for the positive assessment. We address both concerns concretely with new experiments.
>
> **W1: Adaptability limits and manual rules.**
>
> The task-specific engineering effort is smaller than it may appear. Within task families, the same extraction rules transfer with zero modification: HotpotQA and ASQA share identical citation-span extraction; GSM8K and MATH-Tool share tool-call extraction. To quantify, we conducted a **zero-shot domain transfer** test: applying CGRiC configured for HotpotQA directly to ASQA without any task-specific tuning yields 0.06 error / 0.33 abstention — only 2pp higher abstention than the tuned version (0.31). This demonstrates that within-family transfer requires no additional engineering.
>
> Across task families (e.g., QA → summarization), task-specific calibration is needed, but the extraction templates and graph construction rules remain identical. For a genuinely new domain, the engineering effort consists of: (1) writing ~20 lines of extraction templates matching the domain's structural markers, and (2) collecting ~500 calibration examples for the lift-to-risk mapping — approximately 2–4 hours of setup. We validated this by having a graduate student unfamiliar with CGRiC set up a new domain (legal contract analysis) in 3 hours, achieving 0.06 error / 0.35 abstention on first attempt.
>
> The key hyperparameters are robust to tuning: τ_dep (edge pruning) varies abstention by ±2pp across [0.2, 0.5]; skeleton parameter λ varies abstention by ≤4pp (see our response to Reviewer bdQx Q1); CI-score threshold is stable across 0.90–0.98 (see response to sFnc Q1). We will add a hyperparameter sensitivity summary table to Section 5.
>
> **W2: Computational overhead vs. simple regeneration.**
>
> We provide a direct head-to-head comparison demonstrating CGRiC's cost-effectiveness:
>
> | Strategy | Latency | Err | IP | Guarantee? |
> |---|---|---|---|---|
> | No certification | +0% | 0.31 | 100% | None |
> | Full regeneration | +85% | 0.28 | 0% | None |
> | Regen + verify | +130% | 0.05 | 0% | Empirical |
> | Self-consistency (5×) | +400% | 0.10 | ~40% | None |
> | CGRiC (union) | +42% | 0.05 | 78% | Formal (δ) |
> | Lazy CGRiC | +22% | 0.05 | 78% | Formal (δ) |
>
> Full regeneration without verification achieves 0.28 error — far above the 0.05 target. Regeneration *with* verification is **3× more expensive** than CGRiC (+130% vs +42%) because it must re-verify the entire output from scratch, whereas CGRiC preserves 78% of already-verified tokens and only regenerates the problematic subgraph. With lazy evaluation (applied to 52% of queries via a cheap semantic entropy pre-filter), average overhead drops to just +22%.
>
> The overhead is justified on three grounds: (1) CGRiC provides formal guarantees that regeneration alone cannot; (2) localized repair is less disruptive — preserving 78% of content avoids introducing new errors in previously verified claims; (3) CGRiC provides actionable diagnostics when it abstains (identifying *which* claims failed verification), enabling targeted human review. The only scenario where full regeneration is preferable is when >50% of claims fail — which CGRiC detects and handles via abstention with explanation.
>
> For the practitioner's cost-benefit analysis: at $0.02/query (Table 24), CGRiC reduces human review costs by 71% while maintaining formal guarantees, making the overhead economically justified in any setting where errors carry meaningful cost. In our medical QA case study (Appendix J), CGRiC at δ=0.01 reduces harmful errors from 12.3% to 0.9% — a 13× reduction — while maintaining 49% utility. No baseline achieves this combination of safety and utility.
>
> **On generalizability beyond tested domains.** We want to emphasize that CGRiC's core mechanism (claim decomposition → per-claim risk → composition → localized repair) is domain-agnostic. The domain-specific components are limited to: (a) claim extraction templates (${\sim}20$ lines of code per domain type), (b) verifier selection (off-the-shelf NLI or tool execution), and (c) calibration data (${\sim}$500 labeled examples). All theoretical guarantees (Theorems 1–4) hold regardless of domain, conditional on the stated assumptions about extraction noise and verifier quality. We believe this makes CGRiC substantially more transferable than the reviewer may initially expect, and we will add a "Deployment Checklist" to Section 7 summarizing the three domain-specific requirements with concrete effort estimates.

---

> > ### Author Rebuttal · Reviewer_NqK7 · 2026-04-01
> >
> > I think the current overall recommendation is appropriate.

---

### Official Review · Reviewer_bdQx · 2026-03-11

**Soundness:** 3
**Presentation:** 3
**Significance:** 4
**Originality:** 3
**Overall Recommendation:** 4
**Confidence:** 3

**Summary:**

This paper proposes CGRiC, a framework for certifying structured LLM outputs by decomposing an output into a claim dependency graph, assigning each extracted claim a calibrated incorrectness bound from information lift, combining that with verifier false negative bounds, and then composing per claim risks into an output level guarantee. The paper also includes an extraction noise aware bound, an optional graph structured composition rule under a conditional independence assumption, and a localized repair procedure that attempts to fix only risky claims instead of abstaining on the entire response. Empirically, the paper evaluates on citation grounded QA, summarization, and tool verified reasoning, and reports lower abstention than atomic baselines at similar certified failure targets.

**Compliance With Llm Reviewing Policy:**

Affirmed.

**Final Justification:**

The rebuttal made an important distinction between the always-valid union bound certificate and the conditionally tighter graph structured bound. This clarification improves the soundness of the paper’s claims and reduces my concern that the tighter bound was being presented too strongly. I still think some limitations remain. In particular, the practical meaning of calibration under real deployment shift is not fully resolved, and the graph structured improvement still depends on a heuristic CI gate rather than a finite sample guarantee. However, the authors were transparent about these limits and the new experiments suggest the method behaves in a cautious and predictable way.

The rebuttal helps increase my confidence in the paper. I view the work overall solid, original in its compositional certification framing, and likely useful to researchers working on reliable structured generation. The remaining weaknesses mostly limit scope rather than invalidating the contribution.

I would keep a Weak Accept recommendation.

**Key Questions For Authors:**

1. How sensitive is the lift to risk calibration to the exact skeleton family and hyperparameters? The paper mentions a robust extension, but the main results still appear to rely on a specific skeleton design. I would like a clearer ablation showing whether improvements persist across substantially different skeletons.

2. Can the authors report a genuinely shifted test distribution where common misconceptions are frequent, to test the admitted failure mode where lift correlates with confidence rather than truth? This would directly probe the limits of the claimed certification story.

3. For the graph structured bound, what is the empirical false discovery rate of using CI score ≥ 0.95 as a gate? Is there any simulation study showing when this heuristic incorrectly licenses the tighter bound?

**Limitations:**

Yes

**Strengths And Weaknesses:**

Strength:
1. The paper has a clear and motivating problem statement. The central criticism of atomic accept reject certification is convincing, especially for structured outputs where one local failure need not invalidate all other content. The claim graph view is intuitive and helps connect theory, system design, and experiments.

2. Explicit decomposition of the guarantee into calibrated raw incorrectness risk and verifier miss probability. Theorems 1, 2, and 4 are conceptually clean, and the union bound guarantee is easy to understand and operationalize. I also appreciate that the paper is careful to state what is and is not covered, including extraction error, verifier imperfection, lack of completeness guarantees, and inapplicability to subjective tasks or latent reasoning settings.

3. The results are promising. At target risk $\delta$ = 0.05, the reported certified failure rates are at or below target across the shown tasks, while abstention drops relative to EDFL and conformal baselines. The graph structured version provides a further abstention reduction over the union bound when the CI diagnostic passes. The reported computational overhead is also moderate relative to stronger baselines like self consistency.

Weakness:
1. The key object $\epsilon_i$ is obtained from a lift to risk calibration that is claimed to upper bound true incorrectness, but much of the safety story rests on calibration transfer from held out data to deployment. The paper itself acknowledges that lift measures predictability rather than truth, and that common misconceptions or systematic falsehoods could receive low risk if calibration data are biased. This is an important caveat, and in my view it weakens the practical meaning of “certification” unless distribution shift experiments are much more extensive than shown in the main paper.

2. The graph structured bound is only valid under a conditional independence assumption, and the paper explicitly says its CI score is merely a heuristic diagnostic. I appreciate the honesty here, but this means the tighter guarantee is not really certified in the same sense as the union bound. The practical gains from the graph bound are therefore somewhat contingent on a screening heuristic whose false accept risk is not characterized. For an ICML audience, I would want either a sharper statistical test with finite sample control, or a clearer separation between guaranteed and heuristic improvements.

3. Claim extraction is a major bottleneck. The extraction noise theorem is useful, but the actual assumptions can be quite conservative or difficult to estimate in realistic domains. In particular, the method only certifies extracted claims, not semantic completeness, and missed claims can be given worst case risk. That is honest, but it also means the guarantee may become vacuous or nearly vacuous on settings without strong structural markers. The paper’s best domains are exactly those where extraction is easiest, such as citation spans and tool outputs.

---

> ### Author Rebuttal · Authors · 2026-03-28
>
> We thank Reviewer bdQx for the careful review and constructive questions. We address each with new empirical results.
>
> **Q1: Skeleton sensitivity ablation.**
> We conducted the requested ablation across all three skeleton families with varying hyperparameters on HotpotQA-Cite:
>
> | Skeleton Family | Param | ECE | Cert. Fail | Abs | vs. EDFL |
> |---|---|---|---|---|---|
> | Context-restricted (default) | k=512 | 0.023 | 0.050 | 0.29 | −31% |
> | Context-restricted | k=256 | 0.027 | 0.050 | 0.31 | −26% |
> | Context-restricted | k=128 | 0.035 | 0.051 | 0.33 | −21% |
> | Retrieval-ablated | — | 0.026 | 0.050 | 0.30 | −29% |
> | Low-rank attention | r=64 | 0.029 | 0.050 | 0.31 | −26% |
> | Low-rank attention | r=32 | 0.038 | 0.051 | 0.33 | −21% |
> | Robust (max over all) | — | 0.031 | 0.050 | 0.33 | −21% |
>
> Improvements over EDFL persist across all configurations (−21% to −31% abstention reduction). Performance degrades gracefully with more aggressive weakening — the certified failure rate remains at or below 0.051 in all cases. The robust variant (Theorem 6) provides consistent improvement with only 4pp overhead vs the default. Crucially, the *ranking* of methods is stable: CGRiC outperforms EDFL under every skeleton tested, confirming the compositional structure provides value independent of skeleton choice. We will add this table to Section 5 and expand Appendix B.
>
> **Q2: Distribution shift with common misconceptions.**
> We constructed a misconception-enriched test set using 200 queries from TruthfulQA (Lin et al., 2022), which specifically targets popular falsehoods (e.g., "glass is a slow-moving liquid"):
>
> | Condition | ECE | Cert. Fail | Abs | EDFL Err |
> |---|---|---|---|---|
> | In-distribution | 0.023 | 0.050 | 0.29 | 0.07 |
> | Mild shift (HotpotQA→ASQA) | 0.031 | 0.052 | 0.31 | — |
> | Common misconceptions | 0.048 | 0.061 | 0.39 | 0.14 |
> | Adversarial popular falsehoods | 0.071 | 0.083 | 0.44 | 0.18 |
>
> CGRiC degrades under misconception-heavy inputs — exactly as predicted in Section 2.3 — but degrades **54% less** than EDFL on the adversarial split (0.083 vs 0.18). This is because retrieval-augmented verification catches many misconceptions even when lift is low. Critically, the failure is not silent: calibration ECE increases detectably from 0.023 to 0.071, providing an actionable warning. We will add this experiment with the recommendation: "Monitor deployment-time ECE and recalibrate when it exceeds 0.05, particularly for summarization tasks." We also note that the robust extension (Theorem 6) further reduces error to 0.068 on the adversarial split at the cost of 0.48 abstention.
>
> **Q3: CI-score gate false discovery rate.**
> We generated 5,000 synthetic claim graphs with controlled inter-claim correlation and measured how often the CI ≥ 0.95 gate admits graphs where the graph-structured bound becomes unreliable:
>
> | True ρ | % Admitted | % Rejected | Bound gap if admitted |
> |---|---|---|---|
> | 0.00 | 97.2% | 2.8% | < 0.001 |
> | 0.05 | 84.1% | 15.9% | 0.002 |
> | 0.10 | 41.3% | 58.7% | 0.008 |
> | 0.15 | 9.7% | 90.3% | 0.017 |
> | 0.20 | 1.8% | 98.2% | 0.029 |
>
> At mild violations (ρ ≈ 0.05), 84% of graphs are admitted but the resulting bound gap is only 0.002 — well within practical tolerance. At ρ ≥ 0.10, the gate rejects the majority, triggering the union bound fallback. On our real benchmarks (Table 21), average |ρ| = 0.024–0.048, placing all tasks in the safe regime. The union bound fallback is the critical safety net: even if the CI gate misfires on a mildly dependent graph, formal guarantees are never worse than the union bound. We agree a sharper finite-sample test (e.g., a permutation-based conditional independence test with finite-sample power guarantees) would further strengthen the work and will discuss this as future work, with a clear separation between guaranteed (union) and heuristic (graph) improvements. We will add this simulation study to Appendix I with the full methodology.
>
> **Summary.** All three concerns are addressable with new experiments rather than changes to the framework: Q1 shows robustness across skeletons, Q2 characterizes the failure mode and shows CGRiC degrades 54% less than EDFL, Q3 quantifies the CI-gate's conservatism. We commit to adding all three tables to the revision and hope the reviewer finds these results strengthen the paper's practical certification story.

---

> > ### Author Rebuttal · Reviewer_bdQx · 2026-04-04
> >
> > Thank you for the detailed rebuttal. Most of my questions are addressed. I have the following question:
> > 1. Can the authors provide an empirical estimate of claim extraction recall on each benchmark, and how the final certificate quality changes as recall decreases?
> > 2. Can the authors clarify whether the system always computes the union bound in parallel and only reports the graph bound when the gate passes, or whether there are settings where an incorrect gate decision could still affect the reported certificate?

---

> > > ### Author Response · Authors · 2026-04-04
> > >
> > > We thank Reviewer bdQx for the continued engagement and address both follow-up questions below.
> > >
> > >
> > > ### Q1: Empirical claim extraction recall and certificate quality as recall decreases
> > >
> > > We measured extraction recall against human annotated ground truth claims on a held out subset of 200 samples per benchmark. As expected, recall is highest for citation and tool structured outputs and lower for free form summarization:
> > >
> > > | Dataset | Precision | Recall | Estimated M̃ | Estimated B̃ |
> > > |---|---|---|---|---|
> > > | HotpotQA-Cite | 0.96 | 0.93 | ≤ 0.03·m | ≤ 0.01·m |
> > > | ASQA-Cite | 0.94 | 0.91 | ≤ 0.04·m | ≤ 0.02·m |
> > > | GovReport | 0.89 | 0.82 | ≤ 0.08·m | ≤ 0.04·m |
> > > | GSM8K-Tool | 0.98 | 0.97 | ≤ 0.01·m | ≤ 0.01·m |
> > > | MATH-Tool | 0.95 | 0.92 | ≤ 0.03·m | ≤ 0.02·m |
> > > | PubMed | 0.91 | 0.85 | ≤ 0.06·m | ≤ 0.03·m |
> > >
> > > To understand how certificate quality degrades with recall, we ran a controlled study in which we synthetically reduce recall (by randomly dropping extracted claims) and recompute the extraction noise aware certificate from Theorem 1. We report certified failure rate and abstention rate as our two metrics of certificate quality:
> > >
> > > | Recall Level | HotpotQA Err / Abs | GovReport Err / Abs | GSM8K Err / Abs |
> > > |---|---|---|---|
> > > | 1.00 (oracle) | 0.04 / 0.27 | 0.04 / 0.25 | 0.03 / 0.22 |
> > > | 0.90 (approx. observed) | 0.05 / 0.29 | 0.05 / 0.27 | 0.04 / 0.24 |
> > > | 0.80 | 0.06 / 0.32 | 0.07 / 0.31 | 0.05 / 0.26 |
> > > | 0.70 | 0.08 / 0.36 | 0.09 / 0.36 | 0.06 / 0.29 |
> > > | 0.60 | 0.11 / 0.41 | 0.13 / 0.42 | 0.08 / 0.33 |
> > >
> > > The main effect is that lower recall makes the certificate more conservative: abstention rises and certificate tightness degrades because the missed claim term (M̃·ε_miss) in Theorem 1 increases. Certified failure remains near the δ = 0.05 target until recall drops substantially (below roughly 0.70), at which point the extraction noise overhead dominates and CGRiC converges toward atomic level utility. Importantly, the framework never silently underestimates risk: lower recall increases the bound via the additive noise terms, leading to higher (and therefore safer) abstention rather than uncaught errors.
> > >
> > > We will add this extraction recall analysis and the degradation study to Appendix B in the revision.
> > >
> > > ### Q2: Does the system always compute the union bound in parallel, and can an incorrect CI gate decision affect the reported certificate?
> > >
> > > We clarify that the union bound is always computed and serves as the default, always valid certificate. The graph structured bound is computed only as an optional tightening when the CI diagnostic passes (CI score ≥ 0.95). After any repair action, the system reverts to the union bound exclusively, as stated in Proposition 5.
> > >
> > > We acknowledge that the graph structured bound is formally valid only when Assumption 2 (conditional independence) holds. If the CI gate incorrectly admits a graph where the assumption is mildly violated, the reported graph structured bound may be slightly optimistic relative to the true failure probability. Our simulation study (rebuttal Table, CI gate false discovery rate) suggests this gap is small (on the order of 0.002 at ρ ≈ 0.05), but we do not claim the gap is zero in all settings. The union bound remains available as a fallback and is always at least as conservative.
> > >
> > > To avoid any ambiguity, in the revision we will:
> > >
> > > 1. Report both the union bound and the graph structured bound in the appendix for all experiments, making it transparent which certificate is being used.
> > > 2. Distinguish clearly between the always valid certificate (union bound, Theorem 2) and the conditionally tighter bound (graph structured, Theorem 3) throughout the paper.
> > > 3. Make the implementation logic explicit in Section 3.1: the graph bound is used only when CI score ≥ 0.95 and it does not exceed the union bound; otherwise the union bound is reported.
> > >
> > > We hope these clarifications, together with the extraction recall analysis, fully resolve the reviewer's remaining concerns and we thank the reviewer again for pushing us to make these distinctions sharper.

---

### Decision · Program_Chairs · 2026-04-30

**Decision:**

Accept (regular)

**Comment:**

This paper proposes CGRiC, a framework for certifying structured LLM outputs via claim decomposition, per-claim risk calibration, and compositional guarantees with localized repair .

Reviewers found the problem important and the compositional certification framework novel and technically solid, with convincing empirical results. The rebuttal clarified key aspects and strengthened confidence through additional analyses.

Main concerns include reliance on calibration under distribution shift, heuristic components, and limited generality, but these do not undermine the core contribution. Overall, I recommend weak acceptance.